# Linear RAG scanning mediates editing of Igκ variable region repertoires

Xiang Li[1,2,8], Hongli Hu[1,2,4,8], Yiwen Zhang[1,2,5,8], Tammie Zhu[1,2], Ying Guan[1,2], Kai Xu[1,2], Xin Lin[1,2], Camille Hebert[1,2], Himanshu Batra[1,2], Jenny Zhou[1,2], Zhaoqing Ba[1,2,6,7], Duane R. Wesemann[3], Adam Yongxin Ye[1,2✉] & Frederick W. Alt[1,2✉]

V(D)J recombination-mediated Igκ light chain variable region exon assembly in precursor (pre)-B cells involves recombination activating gene (RAG) endonuclease-orchestrated cleavage between and joining of paired Vκ and Jκ gene segments and flanking RAG-targeting recombination signal sequences (RSSs)[1–3]. The 3.1-megabase *Igk* contains 4 Jκs (Jκ1, 2, 4, 5) and 100-plus Vκs in clusters oriented for deletional or inversional joining[2]. Vκ-to-Jκ joining is ordered, with primary Vκ-to-Jκ1 rearrangements occurring first, followed by secondary rearrangements of upstream Vκs that replace primary VκJκ1s by joining to Jκ2-5 (refs. 4,5). Loop extrusion moves deletional-oriented and inversional-oriented, locus-wide Vκs past the Cer/Sis CTCF-binding element-based diffusion platform for short-range diffusional presentation to Jκ1-bound RAG in the primary recombination centre (RC). To achieve diffusion-mediated Vκ-to-Jκ1 joining, *Igk* evolved powerful Vκ-associated and Jκ-associated RSSs[3]. Secondary *Igk* rearrangements replace non-functional or autoreactive primary VκJκ1 rearrangements, expanding the Igκ repertoire and mediating central tolerance by means of receptor editing[4,6–11]. Here we describe studies that elucidate the physiologically critical secondary *Igk* recombination mechanism. Primary deletional and inversional VκJκ1 joins, respectively, delete or displace Cer/Sis, creating a pre-B cell population that harbours secondary VκJκ1-based RCs across the Vκ locus and leaves most unrearranged Vκs immediately upstream of secondary RCs in deletional orientation. High-throughput assays demonstrated that RAG scanning from secondary VκJκ1-based RCs, collectively, extends linearly across the Vκ locus in primary pre-B cell populations. Correspondingly, studies of induced pluripotent stem (iPS) cell-generated mouse models or cell lines with physiological VκJκ-rearrangements further revealed that deletional and, originally, inversional Vκs are mostly captured by Jκ2-5-based secondary RCs in deletional orientation by means of linear RAG scanning. Strong Vκ-RSSs contribute to restricting secondary rearrangements, including potential editing rearrangements, to Vκs immediately upstream of a given secondary RC and support, at a lower level, linear scanning-based inversional Vκ-to-Jκ rearrangements. Our findings implicate Cer/Sis deletion and/or displacement as a developmental switch that converts the two-loop-based diffusional primary *Igk* rearrangement mechanism into a one-loop-based linear scanning secondary rearrangement mechanism.

*Igh* and *Igk* V(D)J recombination is orchestrated by the RAG1–RAG2 heterotetramer bound to a recombination centre (RC) formed around intronic enhancers and J segments of each locus[1,12]. Vs, Ds and Js are flanked by bona fide recombination signal sequences (RSSs), consisting of conserved CACAGTG heptamer, a 12-base pair (bp) or 23-bp spacer, and an AT-rich nonamer that target recombination activating gene (RAG) endonuclease activity[1,12]. RAG cleaves robustly only paired gene segments flanked by bona fide RSSs with, respectively, 12-bp and 23-bp spacers (12RSSs and 23RSSs), which must be properly aligned in parallel in the active sites of the two RAG1 proteins in the RAG complex[13,14]. Cohesin-mediated chromatin loop extrusion contributes to pairing widely separated *Igh* and *Igk* RSSs for V(D)J recombination[3,14–20]. For *Igh*, impeded downstream loop extrusion at the RC leads to continued extrusion of upstream chromatin through the impeded cohesin ring, allowing J_H-23RSS-bound RAG to linearly scan the upstream 2.5 megabases (Mb) of chromatin for D-12RSSs and,

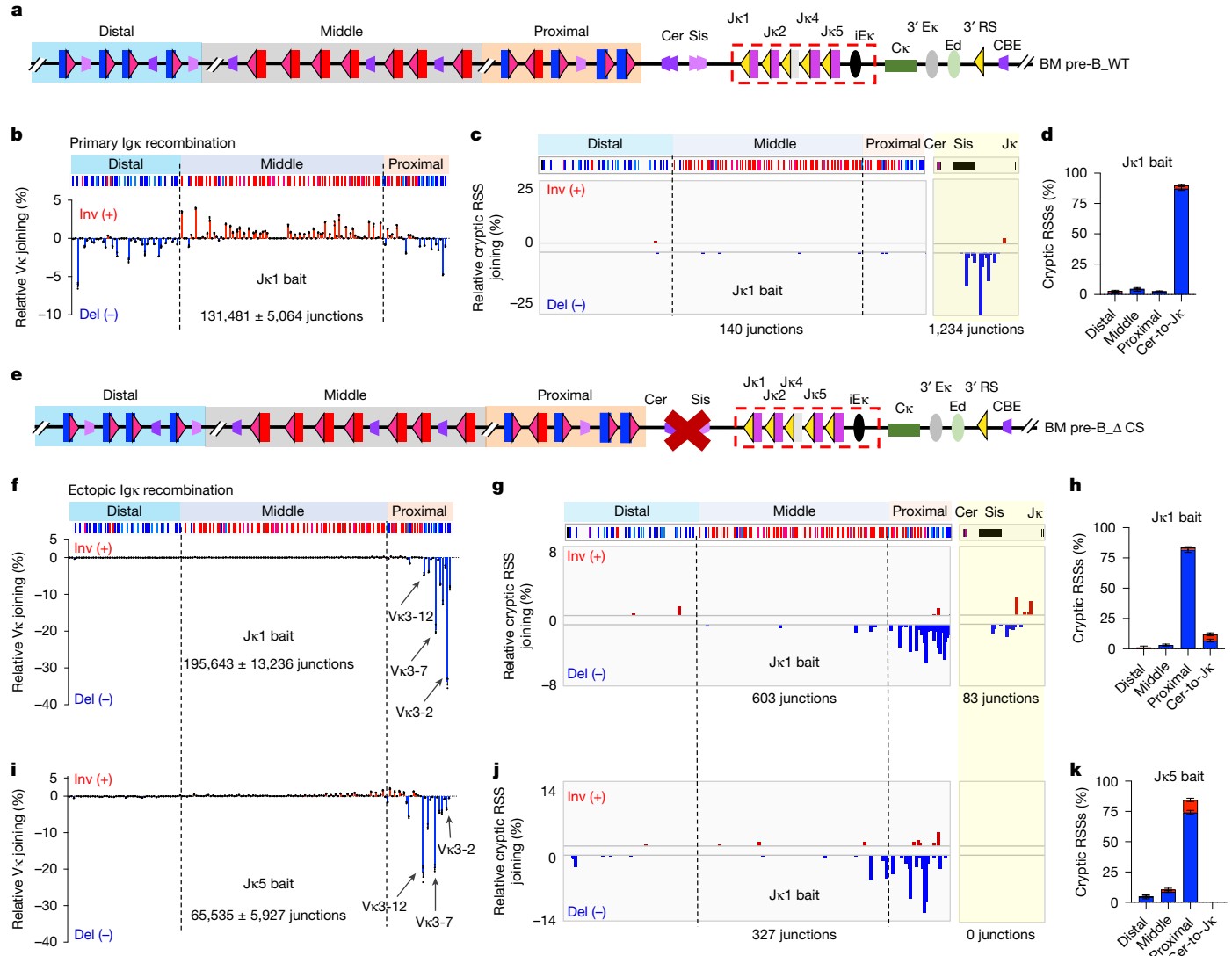

**Fig. 1 | Cer/Sis deletion converts primary rearrangements from short-range diffusion into linear scanning joining mechanism in vivo. a**, Illustration of mouse *Igk* (not to scale). Relative location of distal (blue), middle (grey) and proximal Vκs (orange), CBE-based Cer and Sis, four functional Jκs, *Igk* enhancer (iEκ, 3′ Eκ, Ed) and 3′ RS are indicated. Deletional and inversional Vκs are flanked by 12RSSs (red triangles) and Jκs by 23RSSs (yellow triangles). **b**, Percentage use of individual Vκs in BM pre-B cells baiting from Jκ1. **c**, Percentage of pooled RAG off-target junctions in the Vκ region (left panel) and region between Cer and Jκ (Cer-to-Jκ, right panel) as measured with a Jκ1 bait. **d**, Percentage of inversional (red) and deletional (blue) cryptic RSS junctions within distal, middle and proximal Vκ locus regions, and Cer-to-Jκ region. **e**, Illustration of mouse *Igk* with Cer/Sis deletion (indicated by the red X). **f–h**, Use percentage of individual Vκs (**f**), pooled RAG off-target junctions (**g**) and their quantification (**h**) baiting from Jκ1 in Cer/Sis-deleted mice as in **b–d**. **i–k**, Use percentage of individual Vκs (**i**), pooled RAG off-target junctions (**j**) and their quantification (**k**) baiting from Jκ5 in Cer/Sis-deleted mice as in **b–d**. Relative Vκs-Jκ5 joining is significantly shifted to further upstream within a limited region compared with Vκs-Jκ1 joining in Cer/Sis-deleted mouse (two-sided Welch's *t*-test, *P* = 0.0002). Bona fide and cryptic RSS junctions mapping to positive strand of mouse reference genome are in red and minus strand in blue. Data and error bars in **b,d,f,h,i,k** are presented as mean ± s.e.m. from three biological repeats. WT, wild type.

ultimately, V_H-23RSSs for assembly of complete V_H(D)J_H exons[14–20]. This 'single loop-based' linear scanning mechanism only recognizes complementary RSSs in convergent orientation[15,18–20] (Supplementary Video 1 and Extended Data Fig. 1a). The mouse *Igk* has 4 Jκs with upstream-oriented 23RSSs and more than 100 Vκs with 12RSSs, which lie in three clusters of mostly downstream (deletional) oriented distal Vκs, mostly upstream (inversional) oriented middle Vκs and both downstream and upstream-oriented proximal Vκs[2,3] (Fig. 1a). Vκs across the 3.1 Mb locus are robustly used by the Vκ-to-Jκ1 joining process that generates initial (primary) VκJκ1 repertoires[4]. Vκs in inversional orientation could not be joined by the single loop-based linear RAG-scanning mechanism used by *Igh*. Indeed, for primary Vκ to Jκ1 joining, cohesin-mediated loop extrusion allows a Jκ1-RC-bound RAG to linearly scan just 8 kilobases (kb) upstream to the Sis CTCF-binding

elements (CBEs) of Cer/Sis diffusion platform[3]. Upstream Vκs are moved through a second cohesin ring impeded for downstream extrusion at the Cer CBEs just upstream of Sis[3]. This 'two-loop-based' mechanism allows Vκs to be presented to a RAG-bound Jκ1 RC by short-range diffusion, allowing proper alignment of both deletional and inversional-oriented RSSs for primary Vκ-to-Jκ1 joining[3] (Supplementary Videos 2 and 3 and Extended Data Fig. 1b). To achieve diffusion-based primary joining, *Igk* Vs and Js evolved powerful RSSs[3], compared with those of *Igh* that rely on linked extrusion impediments to enhance their activity[3,14,16,18]. Whereas secondary Vκ rearrangements to Jκs2, 4 and 5 generate a large proportion (roughly 40–60% based on single cell analysis in mouse B cells) of the Igκ repertoire[4,8–10], the mechanism of this physiologically critical process has remained speculative[3].

## Secondary Vκ-to-Jκ joining occurs by RAG scanning

Primary bone marrow (BM) pre-B cells harbouring Cer/Sis deletions generate repertoires skewed towards proximal Vκs[21,22]. In this regard, Cer/Sis deletion may convert the primary Jκ1-based RC into using a mechanism that reflects that of secondary rearrangements. To test this notion, we used high-throughput genome-wide translocation sequencing (HTGTS)-V(D)J-seq[23] to assess the VκJκ repertoire generated from bona fide Jκs in pre-B cells from wild-type mice (Fig. 1a) versus pre-B cells harbouring an 'ectopic' primary *Igk* RC resulting from Cer/Sis deletion (Fig. 1e). HTGTS-V(D)J-seq also detects linear RAG chromatin scanning activity by measuring low-level joins between bona fide Jκ RSSs and prey cryptic RSSs[15,20]. This assay reliably assesses scanning directionality, because direct RAG scanning overwhelmingly uses cryptic RSSs in convergent orientation with the bona fide RSS[15,20]. In normal pre-B cells, primary Jκ1 joins use Vκs across the locus in deletional and inversional orientation, whereas RAG scanning is largely terminated at Sis (Fig. 1a–d). In Cer/Sis-deleted pre-B cells, Jκ1 dominantly uses the very most proximal deletional-oriented Vκs, with the highest use of Vκ3-2 and a lesser extent Vκ3-7, which both have strong RSSs and associated scanning impediments (Fig. 1f and Supplementary Table 1). In these cells, RAG scanning extends through the proximal Vκs, although inversional Vκs in this region are barely used (Fig. 1f–h). To test whether Jκ1 remains dominant for primary rearrangements in the absence of Cer/Sis, we analysed Jκ5 rearrangements. Scanning from Jκ5 also primarily used proximal Vκs, but with relative use significantly shifted upstream to Vκ3-7, Vκ3-10, Vκ3-12 and Vκ3-17, along with, at lower levels, several inversional-oriented Vκs (Fig. 1i). Correspondingly, initiation of RAG scanning from Jκ5 also shifted upstream in the proximal Vκs and into the middle Vκs (Fig. 1j,k). The upstream scanning pattern shift for Jκ5 indicates that Jκ5 scanning initiates from proximal Vκ/Jκ1-RCs and proceeds linearly upstream for a limited distance (diagrammed in Extended Data Fig. 2a–d). Thus, Cer/Sis-deleted pre-B cells still use Jκ1 for primary rearrangements, but the mechanism is converted into one-loop-based linear scanning with limited upstream Vκ use, and lower-level use of inversional-oriented Vκs, all potentially reflecting aspects of the normal secondary rearrangement mechanism (Extended Data Fig. 1c).

To further investigate the mechanistic basis of the dominant usage of Vκ3-2 and Vκ3-7 on Cer/Sis deletion, we performed HTGTS-V(D)J-seq and 3C-HTGTS[16] on wild-type and Cer/Sis-deleted Abelson murine leukaemia virus-transformed pro-B cell lines ('ν-Abl cells'). HTGTS-V(D)J-seq results in RAG-sufficient wild-type and Cer/Sis-deleted ν-Abl lines were very similar to those in pre-B cells (Extended Data Fig. 3a,b,e). On Cer/Sis deletion, Vκ3-2 and Vκ3-7 were dominantly used, and also prominently interacted with the RC, as revealed by 3C-HTGTS in RAG-deficient wild-type and Cer/Sis-deleted ν-Abl lines (Extended Data Fig. 3c,f), probably due to enhanced proximal Vκ transcription[17,24]. In addition, assay for transposase-accessible chromatin using sequencing (ATAC-seq) in these RAG-deficient lines revealed chromatin accessibility of these two highly used proximal Vκs substantially increased, whereas upstream Vκs, which were not used, maintained their varying levels of accessibility (Extended Data Fig. 3d,g). Together, these results indicate both transcription-mediated increased Vκ-RSS RC interactions[17] and Vκ-RSS chromatin accessibility[25–29] may synergistically contribute to the high usage of proximal Vκs during linear RAG scanning on Cer/Sis deletion.

To assay for RAG-scanning activity in the context of both primary and secondary Vκ-to-Jκ recombination in the normal *Igk* locus, we performed HTGTS-V(D)J-seq on DNA from BM pre-B cells isolated from wild-type mice, baiting in separate assays from each of the four functional Jκs (Fig. 2a). Vκ usage patterns appear roughly similar for the different Jκ baits (Fig. 2b–d, compare with Fig. 1b). RAG scanning from the primary Jκ1-RC is largely terminated at Sis (Fig. 1c,d). Most scanning from, Jκ2, 4 and 5 was distributed across the upstream Vκ locus with local orientations mostly corresponding to deletional and inversional Vκ orientations (Fig. 2e–j). However, we did observe a low

level of impeded scanning activity from Jκ2 to Sis (Fig. 2e,h), suggesting that it may also weakly contribute to primary rearrangements. Given the well-established convergent restriction of cryptic RSS usage during scanning[15,20], these findings indicate that RAG-scanning activity from Jκ2-based, 4-based or 5-based secondary RCs mostly occurs after deletion/displacement of Cer/Sis and creation of VκJκ1-based RCs across the Vκ locus through primary Vκ-Jκ1 joining (Extended Data Fig. 4, see legend for details). Specifically, large blocks of RAG-scanning activity mapping in 'inversional' orientation in the reference genome of the middle inversional Vκ domain (Fig. 2e–j) reflect linear RAG scanning from a secondary RC across a portion of the Vκ locus that was inverted by a primary inversional Vκ to Jκ1 recombination event in the middle Vκ domain (Extended Data Fig. 4a and Supplementary Videos 3 and 4). Similarly, predominantly deletional cryptic RSS junctions in the entire distal deletional Vκ domain (Fig. 2e–j) must represent secondary RC-initiated RAG scanning in a largely deleted locus generated by a primary deletional Vκ to Jκ1 recombination event in the distal Vκ domain (Extended Data Fig. 4b). However, RAG-scanning mapping in both orientations in the reference genome of proximal Vκ domain (Fig. 2e–j) can be explained by linear RAG scanning from secondary RCs generated by either primary inversional or deletional Vκ-to-Jκ1 recombination events in the proximal Vκ domain (Extended Data Fig. 4c,d). These findings indicate that limited linear scanning from the numerous secondary VκJκ1-based RCs established during primary joining collectively covers the entire locus. (Extended Data Fig. 4 and legend). Together, our findings provide clear in vivo evidence that *Igk* secondary joining is dominantly mediated by linear RAG scanning.

## Receptor editing uses a restricted Vκ repertoire

To investigate secondary Vκ to Jκ joining in vivo, we generated mouse models for this purpose. We previously described an approach to generate iPS cells from splenic B cells[30]. By screening such iPS cells for physiological VκJκ rearrangements, we identified a 'parental' iPS cell line with a productive Vκ6-23/Jκ5 rearrangement on one *Igk* allele and a non-productive Vκ4-57/Jκ2 rearrangement on the other, with the latter leaving germline Jκ4 and Jκ5 downstream as potential secondary recombination targets (Fig. 3a). The Vκ4-57/Jκ2 non-productive allele actually involved an inversional primary Vκ19-93 to Jκ1 join, followed by a deletional secondary Vκ4-57 to Jκ2 join, leaving numerous originally inversional-oriented middle Vκs in deletional orientation directly upstream of Vκ4-57/Jκ2 and downstream Jκ4 and Jκ5 (Fig. 3a and Extended Data Fig. 5a; see legend for details). We used CRISPR–Cas9-mediated gene editing to mutate the rearranged Vκ6-23/Jκ5 allele into a non-productive rearrangement in a separate ('edited') iPS cell line (Extended Data Fig. 5b). Parental and edited iPS cell lines were used for RAG2-deficient blastocyst complementation[31] to produce chimeric mice in which B and T lineage cells derive from parental or edited iPS cells (Extended Data Fig. 5b).

To assay for potential secondary rearrangements on the Vκ4-57/Jκ2 allele, we isolated BM immature B cells from parental and edited chimeras and performed HTGTS-V(D)J-seq baiting from Jκ4 (Extended Data Fig. 5b). Parental chimeras have a smaller pre-B cell compartment and a larger immature B cell compartment relative to that of edited chimeras, indicating that the pre-existing productive *Igk* rearrangement signals rapid transition from pre-B cells to the next developmental stage[9,10,32] (Extended Data Fig. 5c,d and Supplementary Fig. 1). Even so, we detected very low-level secondary rearrangements in the Vκ4-57/Jκ2 allele in parental chimeras (Fig. 3b). By contrast, chimeras derived from the edited iPS cell line had an 18-fold higher secondary rearrangements level of the Vκ4-57/Jκ2 allele (Fig. 3d,e). Secondary rearrangements in both chimeras used (originally inversional-oriented) deletional-oriented Vκs that were proximal to the secondary RC (Fig. 3c,f), consistent with their use by linear RAG scanning. Few cryptic RSS junctions were recovered from either parental or edited

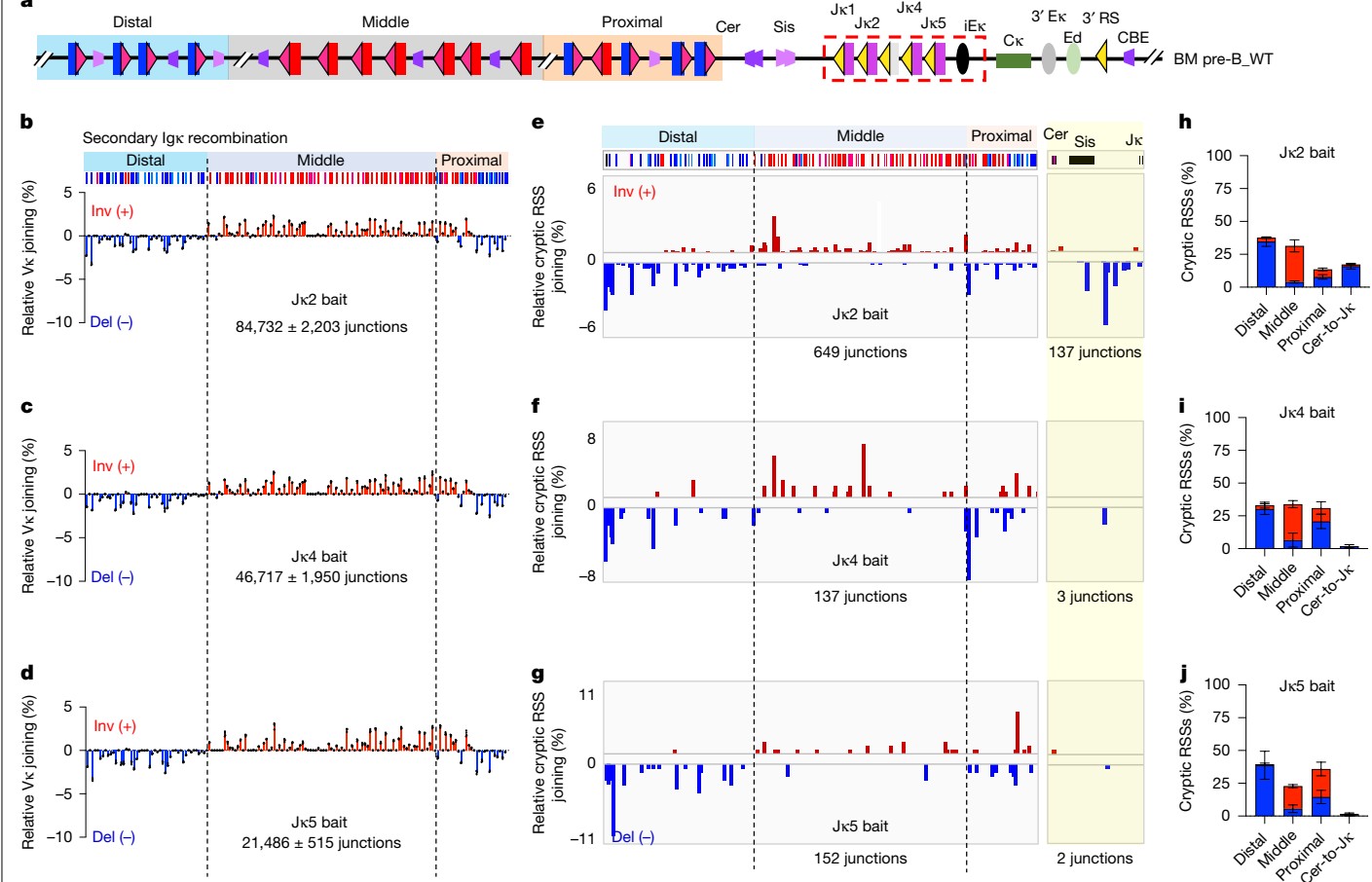

**Fig. 2 | Different mechanisms of *Igk* primary and secondary recombination in vivo. a**, Illustration of mouse *Igk*, as in Fig. 1a. **b**–**d**, Percentage use of individual Vκs in BM pre-B cells baiting from Jκ2 (**b**), Jκ4 (**c**) and Jκ5 (**d**). **e**–**g**, Percentage of pooled RAG off-target junctions in the Vκ region (left panel) and region between Cer and Jκ (Cer-to-Jκ, right panel) as measured with a Jκ2 (**e**), Jκ4 (**f**) and Jκ5 (**g**) in BM pre-B cells. **h**–**j**, Percentage of inversional (red) and deletional (blue) cryptic RSS junctions within distal, middle and proximal Vκ locus regions, and Cer-to-Jκ region as measured with a Jκ2 (**h**), Jκ4 (**i**) and Jκ5 (**j**) in BM pre-B cells. Junction numbers are shown in each panel in this and other figures for comparison of absolute levels. Bona fide and cryptic RSS junctions mapping to positive strand of mouse reference genome are in red and minus strand in blue. Data and error bars in **b**–**d** and **h**–**j** are presented as mean ± s.e.m. from three biological repeats.

chimeras, probably due to selection for productive rearrangements in vivo, especially with only one allele for rearrangement. Consistent with this notion, compared with parental chimeras with a pre-existing functional allele, edited chimeras were highly enriched for productive Vκs/Jκ4 rearrangements (Fig. 3c,f). These data indicate that *Igk* secondary rearrangements in vivo occur by means of linear RAG scanning and use a restricted repertoire of immediately upstream Vκs.

## Strong RSSs dominate secondary deletional joining

The restricted Vκ repertoire used for secondary Vκ-to-Jκ joining in vivo may reflect either confinement of scanning by extrusion impediments or saturation of potential Vκ-to-Jκ rearrangements by robust use of immediately upstream Vκs during linear RAG scanning. G1-arrested *v-Abl* cells undergo Vκ-to-Jκ joining with Vκ usage patterns very similar to those of BM pre-B cells[3,20], providing a genetically tractable system for in depth studies of the secondary *Igk* V(D)J recombination mechanism. To generate *Igk* pre-rearranged *v-Abl* lines poised for secondary rearrangements, we transiently expressed RAG2 in a RAG2-deficient, single *Igk* allele-containing *v-Abl* line[3] in which we mutated the downstream 3′ RS to minimize potential extra-chromosomal rearrangement events (Extended Data Fig. 6a, see legend for details). The downstream 3′ RS has been reported to efficiently join to Vκs and iRS (intronic recombining sequence between Jκ5 and Cκ), which substantially contributes

to *Igk* secondary rearrangements and is particularly critical for κ/λ light chain isotype exclusion[4,10,33,34]. From this modified *v-Abl* line, we isolated a pre-rearranged *v-Abl* line that had joined a middle inversional-oriented Vκ5-37 to Jκ1, which inverted the intervening Cer/Sis-containing sequences and placed nine originally inversional Vκs normally directly downstream of Vκ5-37 in deletional orientation immediately upstream of the secondary Vκ5-37/Jκ1-RC (Fig. 4a and Extended Data Fig. 6a). HTGTS-V(D)J-seq from a Jκ2-bait in this line revealed robust use of three of the five now-deletional orientation Vκs within the roughly 100 kb region upstream of the Vκ5-37/Jκ1-RC (Fig. 4b(ii)). RAG-scanning activity was detected predominantly across this same 100 kb region upstream of the secondary Vκ5-37/Jκ1-RC (Extended Data Fig. 6b,c). These findings confirm linear RAG scanning as the secondary Vκ-to-Jκ joining mechanism in this *v-Abl* line, again with a pattern restricted to immediately upstream Vκs (Fig. 3).

To elucidate the mechanisms that target high use of immediately upstream Vκs during secondary RC-initiated RAG scanning, we used the high-resolution 3C-HTGTS chromatin interaction assay[16] to examine interactions of secondary Vκ5-37/Jκ1-RC with sequences across the upstream *Igk* locus. This study revealed a reproducible interaction peak of the secondary Vκ5-37/Jκ1-RC with the inverted Cer/Sis element 750 kb upstream at the distal end of the inverted Vκ domain; but, by far, the highest peaks of interaction were with two of the five most proximal Vκs (Vκ18-36 and Vκ8-34) (Fig. 4c(ii)). These two Vκs, along

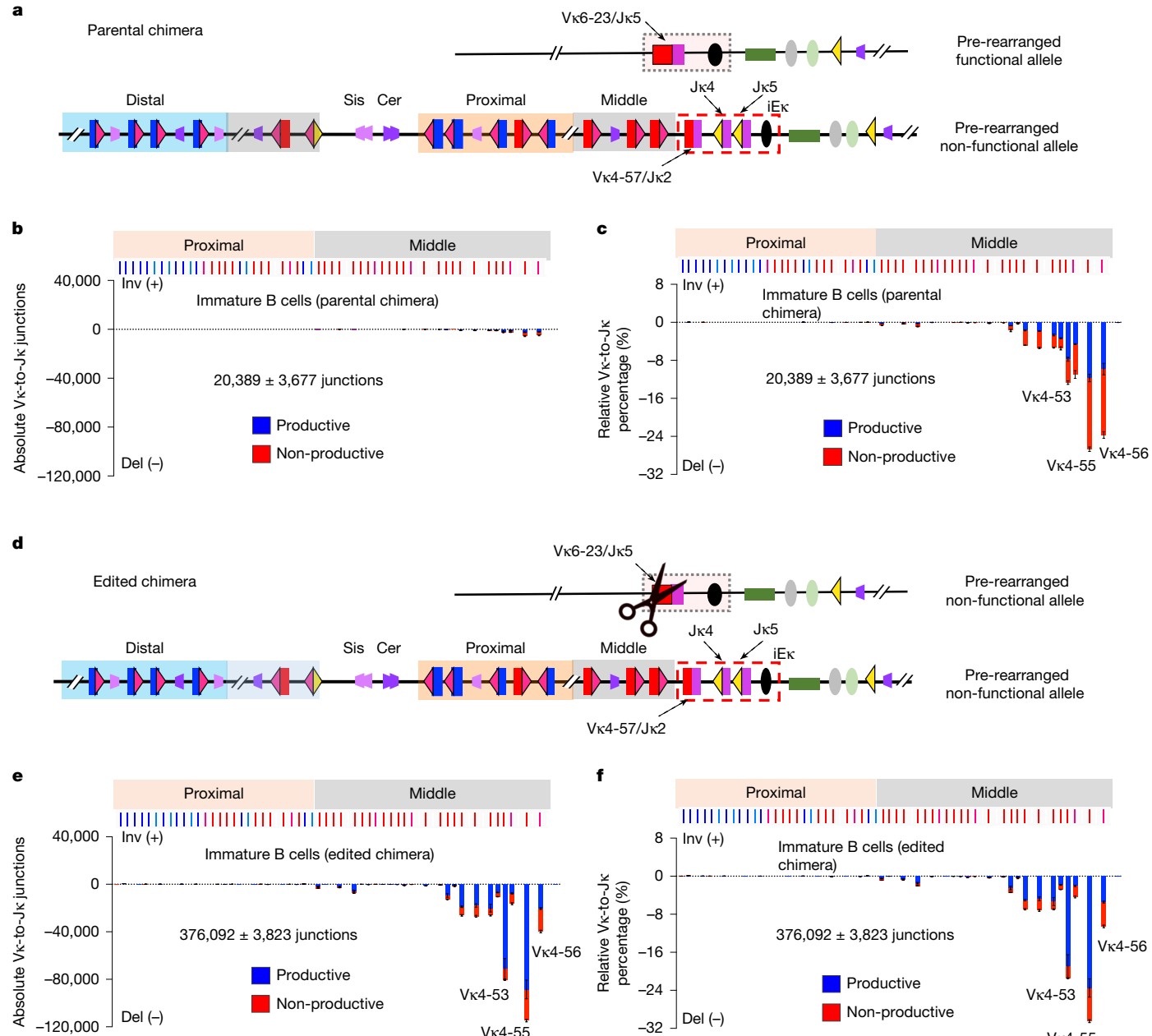

**Fig. 3 | Secondary Vκ-to-Jκ joining uses a restricted Vκ repertoire through linear RAG scanning. a**, Illustration of the *Igk* locus in B cells of parental iPS cell-derived chimeras (parental chimera; not to scale). One *Igk* allele contains only a functional Vκ6-23/Jκ5 rearrangement (pre-rearranged functional allele). The other *Igk* allele contains a non-functional Vκ4-57/Jκ2 rearrangement (pre-rearranged non-functional allele) and downstream germline Jκ4 and Jκ5 as potential secondary recombination targets. **b,c**, Absolute use level (**b**) and percentage (**c**) of individual Vκs, baiting from Jκ4 in DNA from parental chimera BM immature B cells. **d**, Illustration of the *Igk* locus in B cells of edited iPS cell-derived chimeras (edited chimera, not to scale). The functional Vκ6-23/Jκ5 rearrangement was mutated by CRISPR–Cas9 editing to be non-functional (pre-rearranged non-functional allele; see Methods for details). **e,f**, Absolute use level (**e**) and percentage (**f**) of individual Vκs baiting from Jκ4 in DNA from edited chimera BM immature B cells. Productive Vκs/Jκ4 joins are in blue and non-productive joins in red. The percentage of productive rearrangements in edited chimeras compared with that in parental chimeras was significantly increased (two-sided Welch's *t*-test, $P = 0.0011$). Data and error bars are presented as mean ± s.e.m. from three biological repeats.

with the fifth most proximal Vκ (Vκ6-32), which had a lower, but still substantial, interaction (Fig. 4c(ii)), were by far the most highly used Vκs in this line (Fig. 4b(ii)). Notably, all three of these highly used Vκs were substantially transcribed (Fig. 4d(ii)). The two interspersed Vκs that were barely used were not transcribed (Fig. 4d(ii)) and did not have a closely associated RC interaction peak (Fig. 4c; see legend for details). Together these observations suggested that Vκ transcription may act as a physiological linear scanning impediment for *Igk* secondary recombination. To directly test this notion, we deleted the promoter of the highly used Vκ18-36 in the Vκ5-37/Jκ1 pre-rearranged *v-Abl* line,

which markedly reduced Vκ18-36 rearrangement (Fig. 4b(iii)), as well as its transcription and interaction with the secondary RC (Fig. 4c,d(iii)). Notably, the Vκ18-36 promoter deletion also substantially reduced use of a closely linked cryptic RSS hotspot (Extended Data Fig. 6b,c). Vκ18-36 has a relatively weak *Igk* RSS based on recombination information content score[35–37] (Supplementary Table 1) and, correspondingly, is barely used in primary diffusional joining[3]; yet, it was used robustly with the aid of the transcription-based impediment during linear RAG scanning in the secondary rearrangement process (Fig. 4b–d). These findings indicate that use of Vκs with weaker RSSs may be directly

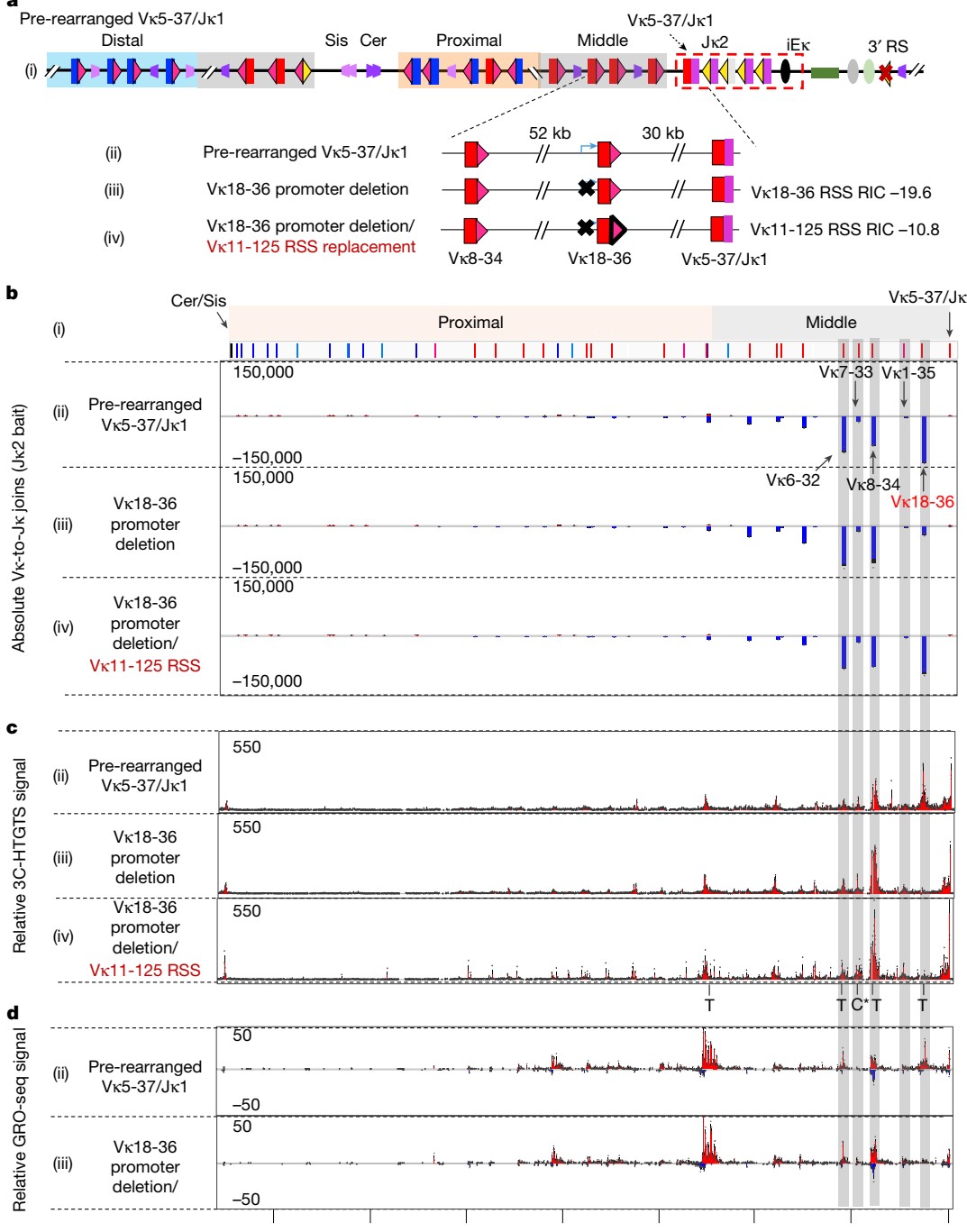

**Fig. 4 | Vκ transcription and strong Vκ RSSs support robust secondary deletional Vκ-to-Jκ joining. a**, (i) Illustration of pre-rearranged Vκ5-37/Jκ1 *v-Abl* line with a primary inversional Vκ5-37/Jκ1 rearrangement that inverted the interval containing Cer/Sis and Vκs, placing a stretch of originally inversional-oriented middle Vκs in deletional orientation directly upstream Jκ2,4,5. (ii) Blow-up of indicated region. (iii) The Vκ18-36 promoter was deleted by CRISPR–Cas9 editing. (iv) The promoter-deleted Vκ18-36-12RSS was replaced with a strong Vκ11-125 12RSS. **b**, (i–iv) Absolute level of Vκs-to-Jκ2 rearrangements in parental pre-rearranged Vκ5-37/Jκ1 line (ii), Vκ18-36 promoter deletion line (iii) and Vκ18-36 promoter deletion line with Vκ11-125 12RSS replacement (iv). Inversional Vκ-to-Jκ2 joins indicated in red and deletional joins in blue. Absolute use of Vκ18-36 is significantly decreased on its promoter deletion (two-sided Welch's *t*-test, $P = 2.87 \times 10^{-6}$). **c**, (ii–iv) 3C-HTGTS profiles of indicated Vκ locus,

baiting from iEκ, in DNA from RAG-deficient pre-rearranged Vκ5-37/Jκ1 line (ii), the Vκ18-36 promoter deletion line (iii) and Vκ18-36 promoter deletion line with Vκ11-125 12RSS replacement (iv). T represents transcription-based interaction peak, and C* represents the CBE-based interaction peak more than 500 bp away from the Vκ7-33 RSS[16]. Relative Vκ18-36-to-RC interaction is significantly decreased on its promoter deletion (two-sided Welch's *t*-test, $P = 0.0093$). **d**, (ii, iii) GRO-seq nascent transcript profiles in the indicated Vκ locus from the RAG-deficient pre-rearranged Vκ5-37/Jκ1 line (ii) or the Vκ18-36 promoter deletion line (iii). Positive strand transcripts are in red and negative strand transcripts in blue. In **b**–**d**, the five most proximal Vκs are highlighted (grey shading). Data and error bars in **b**–**d** are presented as mean + s.e.m. from three biological repeats. chr., chromosome.

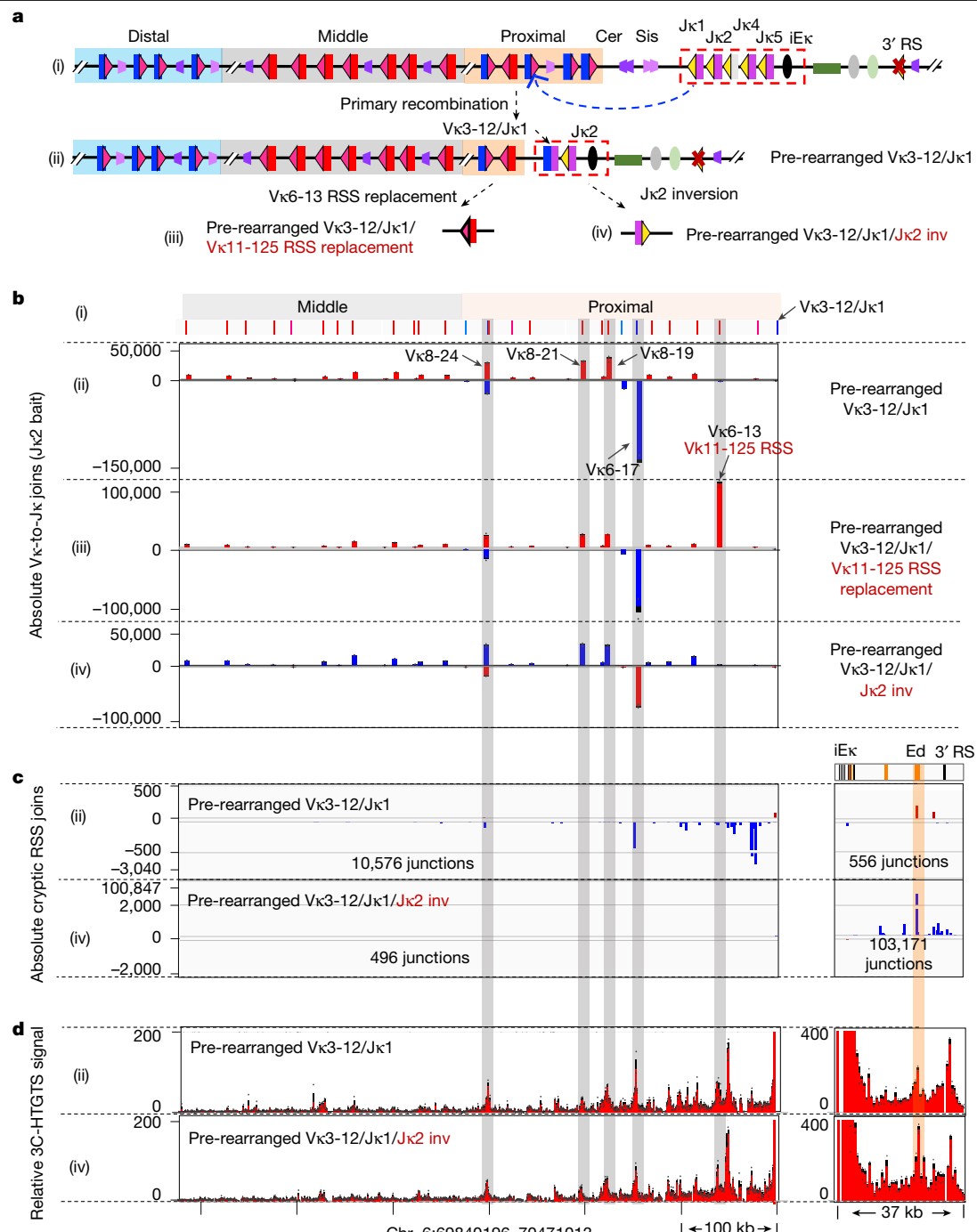

**Fig. 5 | Igk secondary recombination supports inversional Vκ-to-Jκ rearrangements. a**, (i) Illustration of *Igk*. (ii) Illustration of *Igk* pre-rearranged Vκ3-12/Jκ1 *v-Abl* cell line with a primary deletional Vκ3-12/Jκ1 rearrangement that deleted the region containing Cer/Sis and a portion of proximal Vκ domain, placing inversional and deletional Vκs upstream of the secondary Vκ3-12/Jκ1 RC. Jκ3-5 were deleted to eliminate extra-chromosomal joining events. (iii) The inversional Vκ6-13 12RSS was replaced with a strong Vκ11-125 12RSS to generate the Vκ11-125 RSS replacement line. (iv) The Jκ2 and its 12RSS were inverted in the pre-rearranged Vκ3-12/Jκ1 line to generate the Jκ2 inversion line. **b**, (i) Deletional (blue) and inversional (red) Vκs in indicated region are annotated. (ii–iv) Absolute use of individual Vκs, baiting from Jκ2 in DNA from parental pre-rearranged Vκ3-12/Jκ1 line (ii), Vκ11-125 RSS replacement line (iii) and Jκ2 inversion line (iv). Inversional Vκ-to-Jκ2 joins indicated in red and deletional joins in blue. **c**, (ii, iv) Absolute level of pooled RAG off-target junctions in indicated Vκ region (left panel) and RC region (right panel) in pre-rearranged Vκ3-12/Jκ1 line (ii) or Jκ2 inversion line (iv). **d**, (ii, iv) 3C-HTGTS profiles in indicated Vκ region (left panel) and RC region (right panel), baiting from iEκ in DNA from pre-rearranged Vκ3-12/Jκ1 line (ii) or Jκ2 inversion line (iv). Five highly used Vκs in **b**–**d** are highlighted (grey shading), and one highly used cryptic RSS hotspot in **c** and **d** is highlighted (orange shadow). Data and error bars in **b**,**d** presented as mean + s.e.m. from three biological repeats.

enhanced by associated transcriptional impediments that increase their association with the secondary RC and mediate RSS chromatin accessibility during the linear RAG-scanning-based *Igk* secondary recombination.

We further speculated that strong Vκ-RSSs may not require scanning impediments to function during loop extrusion-mediated RAG scanning. To test this notion, we replaced Vκ18-36 RSS with a strong RSS from Vκ11-125 (Supplementary Table 1) in the Vκ18-36

promoter-deleted line (Fig. 4a). Indeed, this strong Vκ RSS alone, in the absence of interaction with secondary RC, supported robust secondary deletional rearrangement through linear RAG scanning (Fig. 4b,c(iv)), indicating strong *Igk* RSSs promote RC-bound RAG-mediated secondary rearrangements during linear scanning. Of note, secondary Vκ-to-Jκ rearrangements during direct RAG scanning are probably saturated, as observed in other experiments described above, as secondary RC-based interactions, done in the absence of RAG, extend much farther than RAG scanning in the Vκ5-37/Jκ1 pre-rearranged *v-Abl* line (Fig. 4b,c and Extended Data Fig. 6b). We conclude that transcription-based scanning impediments and strong Vκ-RSSs cooperate to promote robust use of immediately upstream Vκs and, thereby, restrict available Vκ repertoire during linear RAG-scanning-based secondary *Igk* recombination. Although we cannot rule out a modest role for Vκ locus CBEs, we note that Vκ locus CBEs are less dense and less potent than V$_H$ locus CBEs[3], particularly those of proximal V$_H$s that have a direct role in linear scanning[16] (Extended Data Fig. 1 legend).

## Occurrence of secondary inversional Vκ-to-Jκ joining

Although most physiological primary deletional or inversional Vκ to Jκ1 joining events leave a stretch of Vκs in deletional orientation upstream of the secondary RC (Extended Data Fig. 4a,b), a subset of primary rearrangements place inversional Vκs immediately upstream of the secondary RC (Extended Data Fig. 4c,d). To test whether inversional secondary Vκ-to-Jκ joining occurs, as suggested from the Cer/Sis deletion studies (Fig. 1), we isolated a *v-Abl* line in which Jκ1 was rearranged to a proximal, deletional-oriented Vκ3-12, placing a stretch of inversional Vκs interspersed with 4 deletional Vκs upstream of the secondary RC (Fig. 5a and Extended Data Fig. 7a). We further deleted Jκ3, 4 and 5 and downstream 3′ RS from this line to abrogate confounding extra-chromosomal rearrangements (Extended Data Fig. 7a). In this pre-rearranged Vκ3-12/Jκ1 line, the deletional-oriented Vκ6-17 is the most robustly rearranged; but, at least three inversional Vκs, including Vκ8-24 about 300 kb upstream, are significantly used (Fig. 5b(ii)). HTGTS-V(D)J-seq analyses further revealed that linear RAG scanning also proceeds about 300 kb into the upstream Vκ region (Fig. 5c(ii)). In a second experiment, targeted inversion of immediately upstream deletional Vκs in a Vκ1-117/Jκ1 pre-rearranged *v-Abl* line also led to their relatively robust inversional secondary Vκ-to-Jκ joining during linear RAG scanning (Extended Data Fig. 7b–d). Together, these findings clearly demonstrate that, unlike *Igh*, in which inversional V$_H$-to-DJ$_H$ joining rarely occurs[3,19,20], some Vκs undergo substantial inversional Vκ-to-Jκ rearrangements during linear RAG-scanning-mediated secondary Vκ to Jκ rearrangements.

To test the role of RSS strength in inversional secondary Vκ-to-Jκ rearrangements, we replaced the weak 12RSS of Vκ6-13 (Supplementary Table 1), an inversional Vκ that is roughly 60 kb upstream of secondary RC in the pre-rearranged Vκ3-12/Jκ1 *v-Abl* line, with strong Vκ-12RSSs or weak D$_H$-12RSSs (Extended Data Fig. 8a and Supplementary Table 1). The strong Vκ12-44, Vκ11-125 and Vκ19-93 12RSSs all activated robust inversional Vκ6-13-Jκ2 joining during linear RAG scanning (Fig. 5b(iii) and Extended Data Fig. 8b–d); whereas the weak DFL16.1 and DQ52-upstream RSSs did not (Extended Data Fig. 8e,f). To further elucidate mechanisms promoting inverted Vκ use during linear RAG scanning, we inverted the Jκ2 and its associated RSS in the pre-rearranged Vκ3-12/Jκ1 line to generate the pre-rearranged Vκ3-12/Jκ1/Jκ2-inv line (Fig. 5a). As anticipated, the inverted Jκ2-RSS now directs RAG scanning downstream through the iEκ and Ed to the 3′ RS and 3′ CBE (Fig. 5c(iv) and Extended Data Fig. 9a). On Jκ2 inversion, deletional-oriented Vκ6-17 joined to the inverted Jκ2 through inversion at roughly 50% of the original level, whereas inversional Vκ8-19, Vκ8-21 and Vκ8-24, underwent pseudo-normal deletional joining[38,39] to the inverted Jκ2 (Fig. 5b(iv) and Extended Data Fig. 9b, see legend for details). In this regard, pseudo-normal deletional Vκ-to-Jκ joining occurs robustly in *Igk* primary rearrangements in the

single Jκ5-inverted *v-Abl* line by Cer/Sis-based short-range diffusion[3] (Extended Data Fig. 9c). Given lack of detectable RAG scanning upstream of the RC in the pre-rearranged Vκ3-12/Jκ1/Jκ2-inv line (Fig. 5c(iv)), these findings indicate that both inversional and pseudo-normal deletional Vκ-to-Jκ rearrangements are probably mediated through loop-extrusion-based short-range diffusion to properly align their paired 12/23 strong RSSs in the secondary RC-bound RAG complex. Consistent with this notion, the secondary RC directly interacts with upstream highly used deletional and inversional Vκs regardless of RC-Jκ orientation (Fig. 5d).

## Discussion

*Igk* uses a short-range diffusion-based primary Vκ-to-Jκ1 recombination mechanism to generate a diverse primary VκJκ1 repertoire that uses deletional and inversional-oriented Vκs across the 3.1-Mb-long Vκ locus[3]. We now report that secondary Vκ-to-Jκ2, 4 and 5 rearrangements use an *Igh*-like linear RAG-scanning mechanism (Extended Data Fig. 1c and Supplementary Video 4). Moreover, we find that linear scanning-based secondary rearrangements mostly use a restricted repertoire of immediately upstream blocks of (normally) deletional or inversional-oriented Vκs, with the latter placed in deletional orientation by primary rearrangements (Extended Data Fig. 1c and Supplementary Video 4). Both Vκ transcription-based scanning impediments and, particularly, strong Vκ RSSs enforce such restricted Vκ use by promoting rapid saturation of linear scanning-based secondary rearrangements. Yet, we find that the developing pre-B cell population uses essentially the entire Vκ repertoire for secondary rearrangements, achieving this feat because primary VκJκ1 rearrangements generate secondary RCs across the Vκ locus. *Igk* secondary rearrangements mediate the widely studied receptor editing process[6–10], studies of which were mainly based on mouse models with knocked-in VκJκ rearrangements encoding autoreactive variable regions downstream of Cer/Sis[11,40]. In such models, a non-physiological 'primary-like' RC is created due to presence of Cer/Sis, unknown at the time these models were generated, which would promote sampling of the entire upstream Vκ repertoire through a short-range diffusion, primary joining-like mechanism[3]. B cell-derived iPS cell lines could, in principle, be used for in depth studies of receptor editing when armed with primary VκJκ exons encoding autoreactive or non-pairing Igκ chains and equipped with related or unrelated upstream Vκs. In this regard, Vκs with high sequence similarity are semi-clustered across the Vκ locus[41], which theoretically could promote fine-tuning of the Igκ repertoire by replacing autoreactive Vκs with highly related Vκs harbouring de novo generated CDR3 antigen-binding regions, which might dampen autoreactivity while promoting pairing with the existing IgH chain. In contrast to the *Igh*, which strictly uses deletional-oriented V$_H$s[3,19,20], some inversional Vκs are used during linear scanning-mediated secondary Vκ-to-Jκ rearrangement. This finding implies that the RAG-bound secondary RC has elements that promote diffusion-based synapsis of inversional Vκs with Jκs during linear RAG scanning for joining mediated by their strong RSSs (Extended Data Fig. 1d). Finally, our findings implicate deletion and/or displacement of the Cer/Sis diffusion platform by primary rearrangements as a developmental switch that transforms *Igk* from using the two-loop-domain diffusion-based mechanism into using a one-loop-domain linear scanning-based mechanism (Extended Data Fig. 1 and Supplementary Videos 2–4).

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

[1]Program in Cellular and Molecular Medicine, Howard Hughes Medical Institute, Boston Children's Hospital, Boston, MA, USA. [2]Department of Genetics, Harvard Medical School, Boston, MA, USA. [3]Division of Allergy and Clinical Immunology, Department of Medicine, Brigham and Women's Hospital and Harvard Medical School, Boston, MA, USA. [4]Present address: State Key Laboratory of Immune Response and Immunotherapy, School of Basic Medical Sciences, Division of Life Sciences and Medicine, University of Science and Technology of China, Hefei, China. [5]Present address: Leerink Partners, Boston, MA, USA. [6]Present address: National Institute of Biological Sciences, Beijing, China. [7]Present address: Tsinghua Institute of Multidisciplinary Biomedical Research, Tsinghua University, Beijing, China. [8]These authors contributed equally: Xiang Li, Hongli Hu, Yiwen Zhang. ✉e-mail: yongxin.ye@childrens.harvard.edu; alt@enders.tch.harvard.edu

## Methods

### Experimental procedures

We did not use statistical methods to predetermine sample size. We also did not randomize experiments. During experiments and outcome assessments, investigators were not blinded to allocation.

### Mice

We purchased wild-type 129SV mice from Taconic Biosciences. We have previously described the generation and characterization of 129SV mice harbouring the Cer/Sis deletion[22]. Mouse work was performed in compliance with the relevant ethical regulations established by the Institutional Animal Care and Use Committee of Boston Children's Hospital. The mouse work was performed under protocols approved by the Boston Children's Hospital Institutional Animal Care and Use Committee. Mice were maintained on a 14-h light/10-h dark schedule in a temperature ($22 \pm 3$ °C) and humidity (35% to roughly 70% $\pm$ 5%) controlled environment, with food and water provided ad libitum. Both male and female 4–8-week-old mice were used for experiments. Mouse experiments are not randomized and not blinded to investigators.

### Generation of chimeras from splenic B cell-derived iPS cells

Our laboratory previously generated multiple iPS cell lines reprogramed from splenic B cells, which could contribute to high-grade chimerism in mice[30]. By screening these iPS cell lines using HTGTS-V(D)J-seq assays with baits for each of the functional mouse Jκ coding ends, we determined their immunoglobulin light chain rearrangements. We selected an iPS cell line harbouring, respectively, a productive Vκ6-23/Jκ5 rearrangement on one allele and a non-productive Vκ4-57/Jκ2 rearrangement that left Jκ4 and Jκ5 intact for potential *Igk* secondary recombination on the other allele. We denoted this line as 'parental iPS cell'. The sequences of the Vκ6-23/Jκ5 and Vκ4-57/Jκ2 rearrangements were presented (Supplementary Table 2). We further used HTGTS-V(D)J-seq assay with a Jκ1 RSS-end bait to determine the retained primary Vκ19-93/Jκ1 RSS junction in the Vκ4-57/Jκ2 allele of the parental iPS cell (Supplementary Table 2). To enhance *Igk* secondary rearrangements, we further mutated the productive Vκ6-23/Jκ5 rearrangements by CRISPR–Cas9 editing in the parental iPS cell line using single-guide RNA 1 (sgRNA1) targeting the Vκ6-23/Jκ5 segment (edited iPS cell, Supplementary Table 3). The parental and edited iPS cell lines, which tested negative for mycoplasma, were injected, respectively, into RAG-deficient blastocysts to generate chimeric mice by means of RAG2-deficient blastocyst complementation as described in ref. 31.

### Purification of BM precursor B cells and immature B cells

BM pre-B cells from wild-type or Cer/Sis-deleted 129SV mice were isolated as previously described in ref. 20. Briefly, single cell suspensions were derived from BM of 4–8-week-old wild-type or Cer/Sis-deleted 129SV mice. Then, the suspended cells were staining with anti-B220-APC (eBioscience, no. 17-0452-83, 1:1,000 dilution), anti-CD43-PE (BD Biosciences, 553271, 1:400 dilution) and anti-IgM-FITC (eBioscience, no. 11-5790-81, 1:500 dilution). Finally, B220$^+$CD43$^{low}$IgM$^+$ population was used to determine BM precursor B cells by means of fluorescence-activated cell sorting. As our iPS cell lines contain mixed genetic background of C57BL/6 and 129SV, we used a modified method for our iPS cell-derived chimeras that distinguishes BM progenitor B cells and precursor B cells in C57BL/6 mice[19]. Briefly, single cell suspensions were derived from BM of 4–8-week-old parental and edited chimeras. Then, the suspended cells were staining with anti-B220-BV711 (BioLegend, 103255, 1:300 dilution), anti-CD25-PE (BD Pharmingen, 561065, 1:300 dilution), anti-IgG1-FITC (BD Biosciences, 553443, 1:500 dilution), anti-IgM-APC (Invitrogen, 17-5790-82, 1:500 dilution), anti-CD19-BV421 (BD Biosciences, 562701, 1:300 dilution) and anti-c-Kit-PE/Cy7 (eBioscience, 25-1171-81, 1:300 dilution). Finally, B220$^+$CD19$^+$IgM$^-$IgG1$^-$CD25$^+$ KIT$^-$ and B220$^+$ CD19$^+$ IgM$^-$ IgG1$^+$ populations were used to determine BM precursor B cells and immature B cells, respectively, through fluorescence-activated cell sorting. For DNA extraction and HTGTS-V(D)J-seq assays, we could not get sufficient numbers of BM precursor B cells from the parental chimeras, due to the pre-existing functional Vκ6-23/Jκ5 rearrangement pushing cells through this developmental stage. Therefore, we used BM immature B cells from both the parental and edited chimeras for DNA extraction and HTGTS-V(D)J-seq assays using a Jκ4 coding bait.

### Generation of Jκ1 pre-rearranged *v-Abl* cell lines and derivatives

To generate various Jκ1-pre-rearranged *v-Abl* cell lines, the previously established *Rag2*$^{-/-}$; *Eμ-Bcl2*$^+$ *v-Abl* cell line with single *Igk* allele was used as the initial *v-Abl* cell line[3]. We further mutated the downstream 3′ RS by CRISPR–Cas9 editing to minimize potential extra-chromosomal rearrangement events. RAG2 open reading frame-containing plasmids were transiently transfected into *v-Abl* cell lines by nucleofection. Subsequently STI-571 (Selleck, S2475) was added to the culture medium for 1 day to induce G1-arrest and transient RAG2 expression in the *v-Abl* cell line. STI-571 was removed by changing culture medium and the *v-Abl* cells were cultured for 2 more days before plating single clones in the 96-well plates at a density of roughly 0.3 multiplicity of infection. After culturing for 7 more days, half of cells from each single clone were used for PCR screening that used specific primers for, respectively, germline Jκ1, Jκ2, Jκ4 and Jκ5. Finally, pre-rearranged Vκ1-117/Jκ1, Vκ5-37/Jκ1 and Vκ3-12/Jκ1 *v-Abl* cell lines in which, respectively, Jκ1 was joined to the distal deletional Vκ1-117, the middle inversional Vκ5-37 and the proximal deletional Vκ3-12, were selected and used for subsequent experiments. The sequences of retained Vκ1-117/Jκ1, Vκ5-37/Jκ1 and Vκ3-12/Jκ1 coding junctions in these lines were presented (Supplementary Table 2).

To generate various derivatives of Jκ1-pre-rearranged *v-Abl* cell lines, plasmids that expressing Cas9 and sgRNAs were transfected into the *v-Abl* cells by nucleofection to introduce targeted genetic modifications as previously described[3]. Specifically, sgRNA2 and sgRNA3 that, respectively, target upstream and downstream of Vκ18-36 promoter were transfected into the pre-rearranged Vκ5-37/Jκ1 *v-Abl* cell line to generate its Vκ18-36 promoter-deleted derivative; sgRNA4 and sgRNA5 that, respectively, target upstream of Vκ9-129 and downstream of Vκ1-118 were transfected into the pre-rearranged Vκ1-117/Jκ1 *v-Abl* cell line to generate a Vκs-inverted derivative; sgRNA6 and sgRNA7 that, respectively, target upstream of Jκ3 and downstream of Jκ5 were transfected into the pre-rearranged Vκ3-12/Jκ1 *v-Abl* cell line to delete the DNA segment from upstream of Jκ3 through downstream of Jκ5; sgRNA8 and sgRNA9 that, respectively, target upstream and downstream of Jκ2 were transfected into the pre-rearranged Vκ3-12/Jκ1 *v-Abl* cell line to generate its Jκ2-inverted derivative. All candidate clones with desired genetic modifications were confirmed by Sanger sequencing and presented in Supplementary Table 2. Sequences of all sgRNAs mentioned in this section are listed in Supplementary Table 3. None of the *v-Abl* cell lines was tested for mycoplasma contamination.

### Generation of Cer/Sis-deleted *v-Abl* cell lines

To generate Cer/Sis-deleted *v-Abl* cell lines, plasmids that expressing Cas9 and sgRNAs were transfected into the *Rag2*$^{-/-}$; *Eμ-Bcl2*$^+$ *v-Abl* cell line by nucleofection to introduce targeted genetic modification as previously described in refs. 3,18. Specifically, sgRNA12 and sgRNA13 that, respectively, target upstream of Cer element and downstream of Sis element were transfected into the *Rag2*$^{-/-}$;*Eμ-Bcl2*$^+$ *v-Abl* cell line to delete Cer/Sis elements. The junctional sequence confirming the Cer/Sis-deletion in this line is presented in Supplementary Table 2. Sequences of all sgRNAs mentioned in this section are listed in Supplementary Table 3.

### RSS replacement experiments

RSS replacement experiments were performed using plasmids expressing Cas9 and sgRNA, and short single-stranded DNA oligonucleotides

(ssODN) as donor templates for homology-directed DNA repair as previously described[3]. Briefly, plasmids expressing Cas9 and sgRNA, and ssODN were cotransfected into the Jκ1-pre-rearranged *v-Abl* cell lines. Specifically, sgRNA10 and ssODN1 were used to replace the Vκ18-36 12RSS with a Vκ11-125 12RSS in the pre-rearranged Vκ5-37/Jκ1 *v-Abl* cell line; sgRNA11 and ssODN2–6, respectively, were used to replace the Vκ6-13 12RSS with the 12RSSs from Vκ11-125, DFL16.1-upstream, DQ52-upstream, Vκ12-44 and Vκ19-93 in the pre-rearranged Vκ3-12/Jκ1 *v-Abl* cell line. All candidate clones with desired genetic modifications were confirmed by Sanger sequencing (Supplementary Table 2). Sequences of all sgRNAs and ssODNs mentioned in this section are listed in Supplementary Table 3.

## ATAC-seq and data analyses
ATAC-seq libraries were prepared using the Zymo-Seq ATAC Library Kit (D5458) according to the manual protocol. Briefly, 50,000 living cells from G1-arrest wild-type and Cer/Sis-deleted RAG2-deficient *v-Abl* cells were used for cell lysis and nuclei isolation. The prepared nuclei pellet was incubated with transposition mix containing Tn5 transposase for DNA tagmentation and sequencing adaptor ligation. Finally, ATAC-seq libraries were prepared from adaptor-ligated DNA by PCR amplification and sequenced on Illumina NextSeq2000 using a paired-end 100-bp sequencing kit. ATAC-seq libraries were processed with a previously described pipeline[42].

## HTGTS-V(D)J-seq and data analyses
HTGTS-V(D)J-seq libraries were prepared as described[3]. Briefly, 2 μg of DNA from sorted BM pre-B cells from wild-type and Cer/Sis-deleted mice or BM immature B cells of iPS cell-derived chimeras were used for HTGTS-V(D)J-seq library construction using indicated bait primers listed in figures and figure legends. For RAG2-deficient Jκ1-pre-rearranged *v-Abl* cell lines, RAG2 was reconstituted by means of retroviral infection with the pMSCV-FLAG-RAG2-GFP vector. After adding signal transduction inhibitor to arrest *v-Abl* cells in the G1 stage for 4 days, *v-Abl* cells were collected for DNA purification. Then 10 μg of DNA from G1-arrested RAG-complemented *v-Abl* cells was used for preparing HTGTS-V(D)J-seq libraries using indicated bait primers. The final libraries were sequenced on Illumina NextSeq550 or NextSeq2000 using a paired-end 150-bp sequencing kit. HTGTS-V(D)J-seq libraries were processed with a previously described pipeline[3]. All HTGTS libraries were normalized to 500,000 total reads for comparison. Sequences of bait primers used are listed in Supplementary Table 3.

We developed a mouse *Igk*-specific cryptic RSS usage analysis pipeline. First, cryptic RSS junctions were extracted from HTGTS-V(D)J-seq data using a described pipeline that removes annotated and unannotated bona fide RSSs in the *Igk* locus[3]. Second, for each junction, we annotated deletion length from the RSS and insertion length excluding P-elements of bona fide non-productive Vκ to Jκ rearrangements (to avoid influences of selection) in mouse BM pre-B cells and *v-Abl* cell lines. This analysis revealed essentially no bona fide RSS-associated junctions with greater than 5 bp deletions in Jκ coding bait sequences or greater than 7 bp deletions in Vκ coding prey sequences, and only 1–2% with greater than 2 bp insertions between Vκ-to-Jκ junctions. Accordingly, we filtered out cryptic RSS junctions containing deletions greater than 5 bp in bait, greater than 7 bp deletions in prey, or greater than 2 bp junctional insertions. Third, to exclude misaligned reads, we filtered out cryptic RSS junctions containing more than 2 mutations within 20 bp of the cryptic RSS-associated surrogate coding prey sequences. The programs for this mouse *Igk*-specific cryptic RSS usage analysis along with annotated data on which it is based have been added to the GitHub repository (https://github.com/Yyx2626/HTGTS_related/tree/main/Igk_specific_anno_and_filter). We note that this pipeline should not be used for off-target analysis of cryptic RSS junctions in human pre-B cells because, unlike mouse pre-B cells, human pre-B cells express terminal deoxynucleotidyl transferase and add N-regions[22,38].

## 3C-HTGTS and data analyses
3C-HTGTS libraries using an iEκ bait were prepared from the G1-arrested RAG2-deficient *v-Abl* cells as described in ref. 3. Final 3C-HTGTS libraries were sequenced on Illumina NextSeq550 or NextSeq2000 using a paired-end 150-bp sequencing kit, and were processed with a previously described pipeline[3] (for details, see 'Code availability'). All 3C-HTGTS libraries were normalized to 65,488 total junctions, which represents the smallest recovered number of junctions among the 3C-HTGTS libraries that were compared. Primer sequences used for generating 3C-HTGTS libraries are listed in Supplementary Table 3.

## GRO-seq and data analyses
Global run-on sequencing (GRO-seq) libraries were prepared from the G1-arrested, RAG2-deficient *v-Abl* cell lines as described in ref. 20, sequenced using Illumina NextSeq550 or NextSeq2000 using paired-end 150-bp sequencing kits and processed with a previously described pipeline in ref. 20.

## Quantification, statistics and reproducibility
All HTGTS-V(D)J-seq, 3C-HTGTS, ATAC-seq and GRO-seq experiments have been repeated three times independently with similar results. NextSeq550 control software (v.2.2.0) and NextSeq 1000/2000 control software (v.1.5.0.42699) were used for high-throughput sequencing data collection. Graphs were generated using GraphPad Prism 9, IGV (v.2.22.1) and R version 3.6.1. FlowJo (v.9.3.2) was used for analysing the fluorescence-activated cell sorting data. After normalization, HTGTS-V(D)J-seq, 3C-HTGTS, ATAC-seq and GRO-seq signals of three repeats were merged as average ± s.e.m. of the maximum value. An unpaired, two-sided Welch's *t*-test was used to determine the statistical significance of difference between indicated samples, with *P* values presented in relevant figure legends.

## Availability of materials
All plasmids, cell lines and mouse lines used in this study are available from the authors upon request.

## Reporting summary
Further information on research design is available in the Nature Portfolio Reporting Summary linked to this article.

## Data availability
HTGTS-V(D)J-Seq, 3C-HTGTS and GRO-seq sequencing data reported in this study have been deposited in the ArrayExpress database under the accession number E-MTAB-16001 for HTGTS-V(D)J-Seq data, E-MTAB-16007 for 3C-HTGTS data and E-MTAB-16014 for GRO-seq data. ATAC-seq sequencing data reported in this study have been deposited in the ArrayExpress database under the accession number E-MTAB-16602. Source data are provided with this paper.

## Code availability
HTGTS-V(D)J-seq and 3C-HTGTS data were processed through published pipelines as described[3]. Data generated from NextSeq550 or NextSeq2000 were demultiplexed through TranslocPreprocess.pl, a pipeline available through GitHub at http://robinmeyers.github.io/transloc_pipeline/. The pipeline for analysing HTGTS-V(D)J-seq data is available through GitHub at http://robinmeyers.github.io/transloc_pipeline/. The pipeline for analysing 3C-HTGTS data is available through GitHub at https://github.com/Yyx2626/HTGTS_related. The pipeline for GRO-seq data analysis is available through GitHub at https://github.com/Yyx2626/Fred_Alt_Lab/tree/master/GROseq. The pipeline for ATAC-seq data analysis is available through GitHub at https://github.com/nf-core/atacseq. The mouse *Igk*-specific cryptic RSS usage analysis

pipeline is available through GitHub at https://github.com/Yyx2626/
HTGTS_related/tree/main/Igk_specific_anno_and_filter.

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

**Acknowledgements** We thank Alt laboratory members for contributions to the study, particularly H.-L. Cheng for advice with embryonic stem cell culture, A. Williams, B. de Oliveria and C. Pamplona for help of blastocyst injection and mouse care. This work was supported by National Institutes of Health grant no. R01 AI020047 (to F.W.A.). X. Li is supported and H.H. was supported by Cancer Research Institute Irvington Postdoctoral Fellowship (grant nos. CRI5278 to X. Li and CRI5352 and CRI4203 to H.H.). F.W.A. is an investigator of the Howard Hughes Medical Institute.

**Author contributions** F.W.A. and X. Li designed the overall study with help from H.H. and Y.Z. X. Li, H.H. and Y.Z. performed most of the experiments with the help of T.Z. and C.H. X. Li, Y.G. and K.X. performed 3C-HTGTS experiments in the Jκ1-pre-rearranged *v-Abl* cells. X. Li, H.B. and J.Z. performed HTGTS experiments in the Cer/Sis-deleted mice. X. Lin applied bioinformatics pipelines for data analysis. A.Y.Y. designed and applied the new mouse *Igk*-specific bioinformatics pipeline and also did statistical analyses. D.R.W. generated the iPS cells from splenic B cells. Z.B. performed preliminary experiments helpful for informing these studies and generated Cer/Sis-deleted *v-Abl* cells. X. Li, H.H., Y.Z. and F.W.A. analysed and interpreted data. F.W.A., X. Li and A.Y.Y. designed the figures. F.W.A. and X. Li wrote the paper and other authors helped polish the paper.

**Competing interests** The authors declare no competing interests.

**Additional information**
**Correspondence and requests for materials** should be addressed to Adam Yongxin Ye or Frederick W. Alt.

### a. One-loop-based scanning mechanism for IgH V(D)J rearrangements

### b. Two-loop-based diffusional joining mechanism for Igκ primary rearrangements

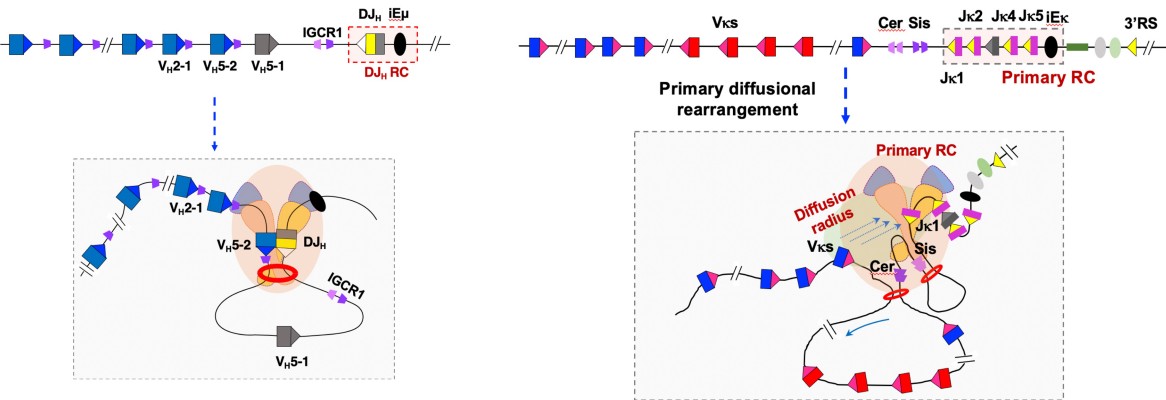

### c. One-loop-based scanning mechanism for Igκ secondary deletional rearrangements

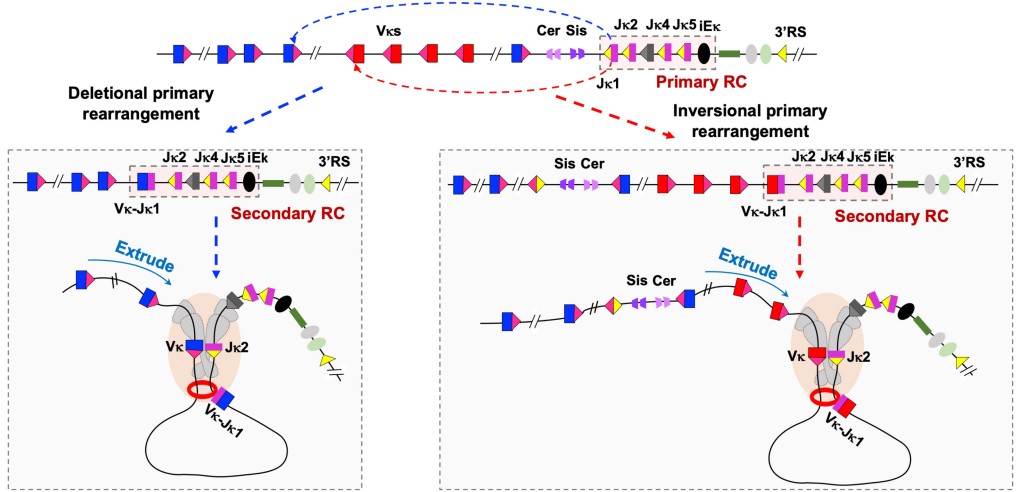

### d. One-loop-based scanning mechanism for Igκ secondary inversional rearrangements

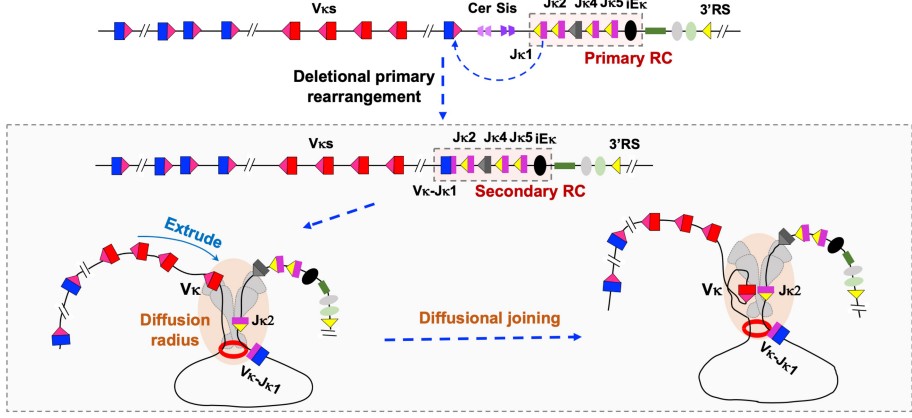

**Extended Data Fig. 1** | See next page for caption.

**Extended Data Fig. 1 | One-loop scanning-based versus two-loop diffusion-based mechanisms for *Igh* and *Igk* V(D)J recombination. (Related to Figs. 1–5). a. One-loop-based linear RAG scanning model for V$_H$-to-DJ$_H$ rearrangements**. Active chromatin of the transcribed nascent *Igh* RC serves as a dynamic barrier to impede downstream cohesin-mediated loop extrusion and recruit RAG, which binds a J$_H$RSS to form an active *Igh* RC[14,16]. This one-loop J$_H$-based *Igh* RC programs RAG to linearly scan the upstream D-containing 100 kb domain from the RC to the IGCR1 CBE-based loop anchor, only using the downstream D-12RSSs in convergent orientation with the J$_H$-23RSS, despite compatibility of upstream D-12RSSs for joining[14,17] (Supplemental Video 1). Joining establishes a new D-J$_H$ RC[16]. Upon WAPL down-regulation, which partially neutralizes IGCR1 CBE-based scanning impediments[19,20,43], scanning moves upstream chromatin from the DJ$_H$ RC to a block of proximal V$_H$s with strong CBEs that lie within 20 bp of their RSSs[16]. Due to weak *Igh* RSSs, proximal V$_H$s require RSS-associated CBEs to promote interactions with the RC for rearrangement[16]. As WAPL down-regulation partially neutralizes proximal V$_H$-RSS CBEs, scanning proceeds into the distal V$_H$s[20,43]. Distal V$_H$s lack closely associated CBEs but are mostly transcribed[16,18,20]. Transcription, a well-known dynamic loop extrusion impediment[17,44–46], facilities their interaction with the RC and their rearrangement[17]. During one-loop linear RAG scanning, only *bona fide* D or V$_H$ RSSs in convergent orientation are robustly used due to the weakness of *Igh* RSSs[3,17,20]. **b. Two-loop-based diffusional model for primary Vκ-to-Jκ1 rearrangements**. Cer/Sis CBEs are more potent than those of IGCR1, which generates a platform for diffusional access of Vκs as they are moved past the closely associated Jκ1-based RAG-bound primary RC[3]. Such long-range extrusion of the Vκ locus can happen even with very high WAPL-levels in *v-Abl* cells, because CBEs in the Vκ locus are less dense and less potent than those in V$_H$ locus[3]. For primary diffusion-based Vκ-to-Jκ1 rearrangements, CBEs and transcription, and other scanning impediments may enhance transient interactions with strong Cer CBEs that help promote transient diffusion-based access of Vκs to the primary Jκ1-RC[3]. For this process, very strong Vκ and Jκ RSSs are required to promote joining[3]. **c. One-loop-based linear RAG scanning model for secondary deletional and (previously) inversional Vκ joining**. Primary Vκ to Jκ1 rearrangements delete or displace Cer/Sis creating VκJκ1-based secondary RCs at every rearranged Vκ across the locus in a population of

pre-B cells. Most deletional and inversional Vκ to Jκ1 joins also leave blocks of deletional- or previously inversional-oriented Vκs, in deletional orientation upstream of secondary VκJκ1-RCs that employ Jκs2,4,and 5[4,5,10]. Because of Cer-Sis deletion/displacement, the *Igk* secondary VκJκ1-RC serves as a dynamic loop anchor that employs direct, one-loop based, linear RAG scanning similar to that used by *Igh* for V(D)J recombination. Notably, besides having less potent CBEs than *Igh*, none of the Vκs have RSS-associated CBEs that would promote their direct RC interaction[3,47]. Finally, direct scanning-based *Igk* secondary rearrangements, like Igκ primary rearrangements, occur in the presence of robust WAPL levels that block nearly all V$_H$ rearrangements in the same cells[3,20]. While there is no evidence that rules out a role for CBEs in promoting *Igk* secondary rearrangements, there is no evidence that directly supports such a role. However, our current findings implicate Vκ transcription in increasing association with the RC[17] and also in mediating its chromatin accessibility[25–29], both of which, coupled with strong Vκ RSSs[3], likely lead to saturation of secondary rearrangements in a limited distance upstream of the RC. We note that the retained upstream CBE-based Cer/Sis elements could also contribute to limiting RAG-scanning upstream from some secondary RCs established by inversional primary rearrangements. **d. One-loop-based scanning mechanism for *Igk* secondary inversional rearrangements**. Our current findings indicate, in contrast to linear scanning in the *Igh*, inversional Vκs, can join, at reduced frequency to deletional-oriented Vκs during direct one-loop-based scanning through the secondary RC. While such utilization is mediated by strong *Igk* RSSs, such inverted Vκ joining likely involves a short-range diffusional aspect for proper pairing. The mechanism for such a process remains to be elucidated. In theory, more distant Vκ elements such as CBEs might contribute. **Evolutionary Implication of these findings**. Human *IGK* and mouse *Igk* have conserved organization and key regulatory elements, including Cer/Sis-like elements and predicted strong RSSs[3], indicating potentially conserved *Igk* two-loop-based primary and one-loop-based secondary rearrangements mechanisms. Also, such one-loop versus two-loop joining mechanisms could also potentially contribute to other immune receptor loci that undergo just V to J rearrangements and have frequent editing, including the Igλ locus (*IGL*) in human and TCRα locus in human (*TRA*) and mouse (*Tra*)[48–50].

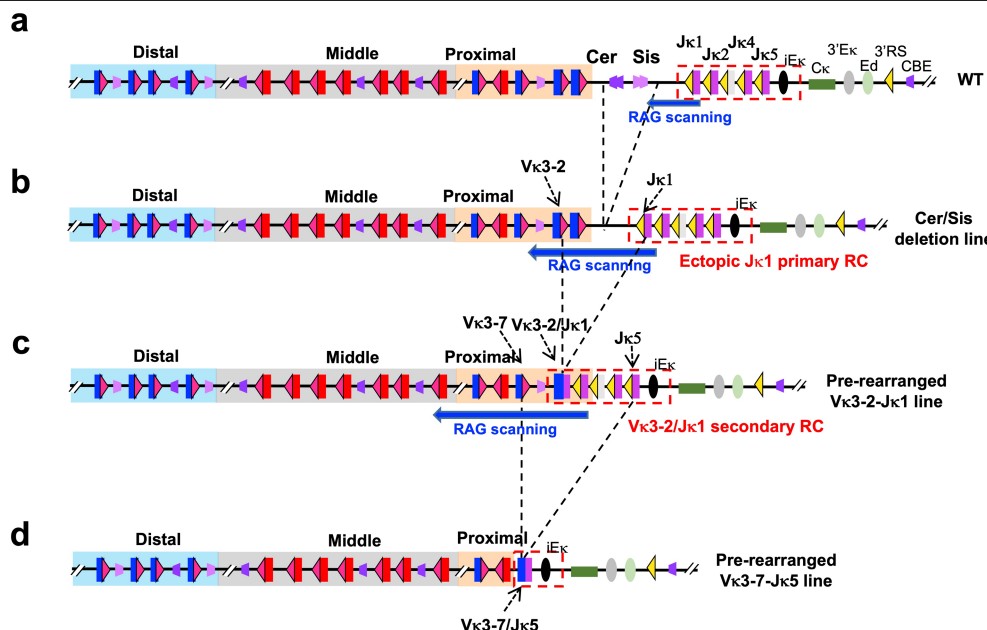

**Extended Data Fig. 2 | Illustration of Vκ-Jκ rearrangements in Cer/Sis-deleted BM pre-B cells. Related to Fig. 1. a.** Illustration of WT *Igk* locus (not to scale). **b.** Illustration of Cer/Sis-deleted *Igk* locus that Cer/Sis is deleted by CRISPR/Cas9-mediated editing to creates an ectopic RC for Jκ1-based primary rearrangement. **c.** Illustration of production of pre-rearranged Vκ3-2/Jκ1 line, in which Jκ1 joins to a proximal deletional Vκ3-2 via linear RAG scanning from the ectopic primary Jκ1-RC. This rearrangement also creates a new Vκ3-2/Jκ1-RC for secondary rearrangements. **d.** Illustration of production of pre-rearranged Vκ-Jκ5 line, in which Jκ5 joins to a proximal deletional Vκ3-7 via linear RAG scanning from the secondary Vκ3-2/Jκ1-RC. These illustrations show an example that, in Cer/Sis-deleted BM pre-B cells, RAG scanning from an ectopic primary Jκ1-RC is limited to the proximal Vκ locus and creates Vκ/Jκ1 secondary RCs that are also restricted in the proximal Vκ locus. Thus, secondary rearrangements from other Jκs, including Jκ5, linearly extend from the proximal Vκ/Jκ1-RCs only for a limited distance upstream before being saturation.

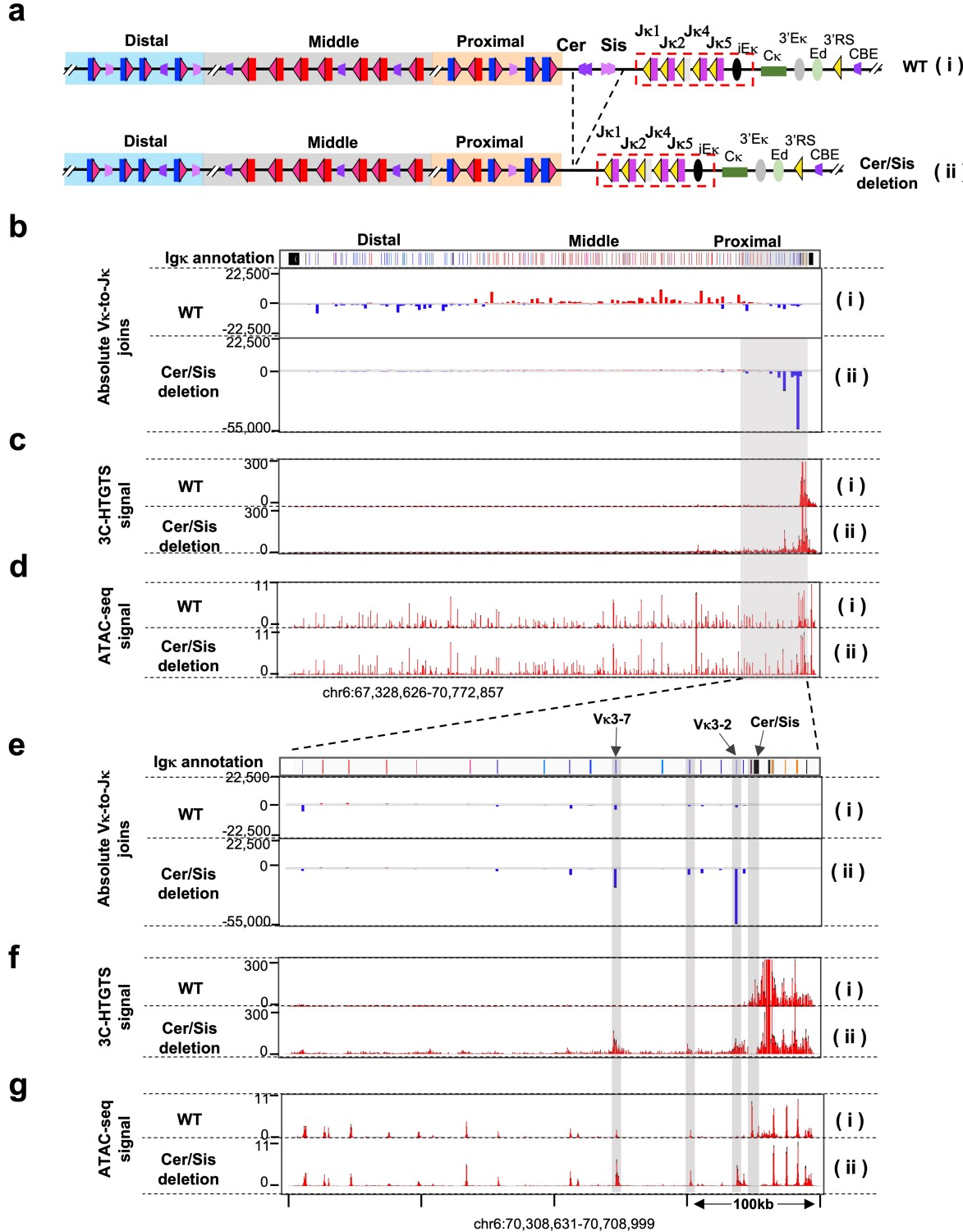

**Extended Data Fig. 3** | See next page for caption.

**Extended Data Fig. 3 | Cer/Sis deletion enhances proximal Vκs usage, interaction with RC and chromosomal accessibility in v-*Abl* cells. Related to Fig. 1. a**. Illustration of Cer/Sis deletion in the RAG-deficient *v-Abl* cells. (i) Illustration of WT *v-Abl* cells. (ii) Illustration of Cer/Sis-deleted *v-Abl* cells. **b**. (i-ii) Absolute utilization of individual Vκs, baiting from Jκ1 in DNA from WT *v-Abl* line (i) and Cer/Sis-deleted *v-Abl* line (ii) with complementation of RAG. Inversional joins are in red and deletional joins in blue. **c**. (i-ii) 3C-HTGTS profiles of *Igk* locus, baiting from iEκ in DNA from RAG-deficient WT *v-Abl* line (i) and Cer/Sis-deleted *v-Abl* line (ii). **d**. (i-ii) ATAC-seq profiles in the *Igk* locus in DNA from RAG-deficient WT *v-Abl* line (i) and Cer/Sis-deleted *v-Abl* line (ii). **e**. (i-ii) Absolute utilization of indicated proximal Vκs, baiting from Jκ1 in DNA from WT *v-Abl* line (i) and Cer/Sis-deleted *v-Abl* line (ii) with complementation of RAG. Inversional joins are in red and deletional joins in blue. **f**. (i-ii) 3C-HTGTS profiles of indicated proximal Vκ and RC locus, baiting from iEκ in DNA from RAG-deficient WT *v-Abl* line (i) and Cer/Sis-deleted *v-Abl* line (ii).

**d**. (i-ii) ATAC-seq profiles in the indicated proximal Vκ and RC locus in DNA from RAG-deficient WT *v-Abl* line (i) and Cer/Sis-deleted *v-Abl* line (ii). Data and error bars in **b-g** are presented as mean + s.e.m from three biological repeats. Absolute utilization of most proximal deletional Vκs, from Vκ3-1 to Vκ3-12, is significantly increased upon Cer/Sis deletion (two-sided Welch's t-test, $P = 0.0011$), while those of other upstream Vκs is significantly decreased upon Cer/Sis deletion (two-sided Welch's t-test, $P = 0.0002$). RC interactions of most proximal deletional Vκs, from Vκ3-1 to Vκ3-12, is significantly increased upon Cer/Sis deletion (two-sided Welch's t-test, $P = 0.0097$), while those of other upstream Vκs remain unchanged upon Cer/Sis deletion (two-sided Welch's t-test, $P = 0.6359$). ATAC signal of most proximal deletional Vκs, from Vκ3-1 to Vκ3-12, is significantly increased upon Cer/Sis deletion (two-sided Welch's t-test, $P = 0.0329$), while those of other upstream Vκs remain unchanged upon Cer/Sis deletion (two-sided Welch's t-test, $P = 0.3256$).

## a

### Primary RC-Jκ1 joins to middle inversional Vκ

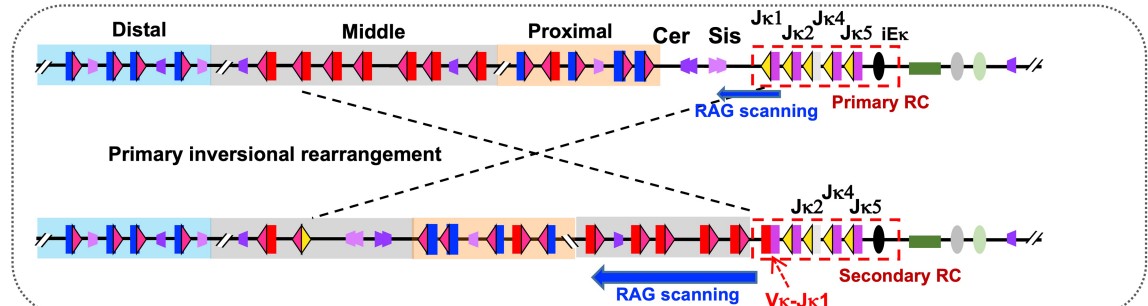

## b

### Primary RC-Jκ1 joins to distal deletional Vκ

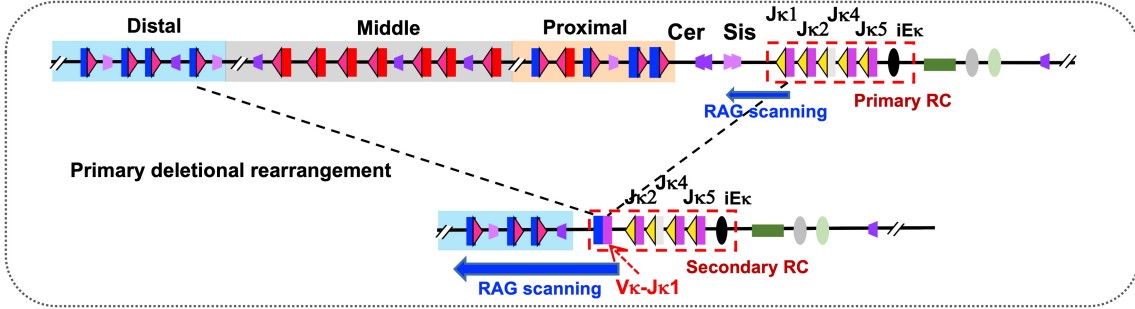

## c

### Primary RC-Jκ1 joins to proximal inversional Vκ

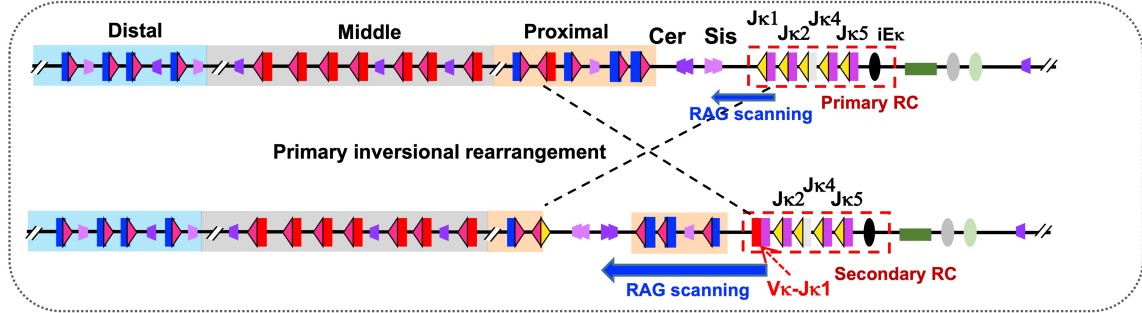

## d

### Primary RC-Jκ1 joins to proximal deletional Vκ

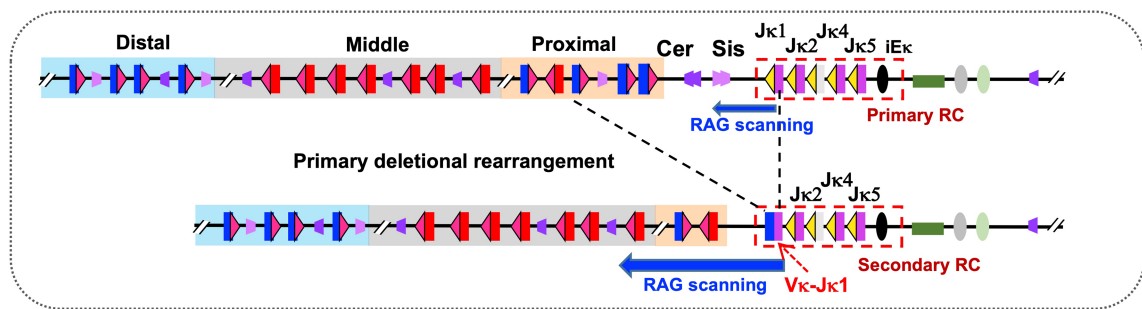

**Extended Data Fig. 4** | See next page for caption.

**Extended Data Fig. 4 | RAG scanning-mediated *Igk* secondary recombination in WT BM pre-B cells. Related to Fig. 2. a**. Illustration of RAG scanning-mediated *Igk* secondary recombination when primary Jκ1 joins to a middle inversional Vκ that inverts the interval containing Cer/Sis, middle and proximal Vκs (not to scale). **b**. Illustration of RAG scanning-mediated *Igk* secondary recombination when primary Jκ1 joins to a distal deletional Vκ that deletes the interval containing Cer/Sis, distal, middle and proximal Vκs. **c**. Illustration of RAG scanning-mediated *Igk* secondary recombination when primary Jκ1 joins to a proximal inversional Vκ that inverts the interval containing Cer/Sis and proximal Vκs. **d**. Illustration of RAG scanning-mediated *Igk* secondary recombination when primary Jκ1 joins to a proximal deletional Vκ that deletes the interval containing Cer/Sis and proximal Vκs. In WT BM pre-B cells, primary Jκ1 joining to the distal Vκs and a majority of middle inversional Vκs places a stretch of Vκs in deletional orientation directly upstream secondary RC. However, Jκ1 joining to proximal Vκs and a short proximal portion of the middle inversional Vκs can place Vκs in both deletional and inversional orientations immediately upstream secondary RC, which complicates interpretation of the scanning patterns in this relatively short Vκ region. In this regard, in WT BM pre-B cells, primary Jκ1 rearrangements utilize the entire Vκ repertoire via a short-range diffusional mechanism, which create Vκ/Jκ1-RCs across the Vκ locus. Although RAG scanning for secondary rearrangements is restricted for a limited distance, for population of BM pre-B cells, RAG scanning from these numerous Vκ-Jκ1 RCs across the locus collectively cover the entire Vκ locus for secondary rearrangements.

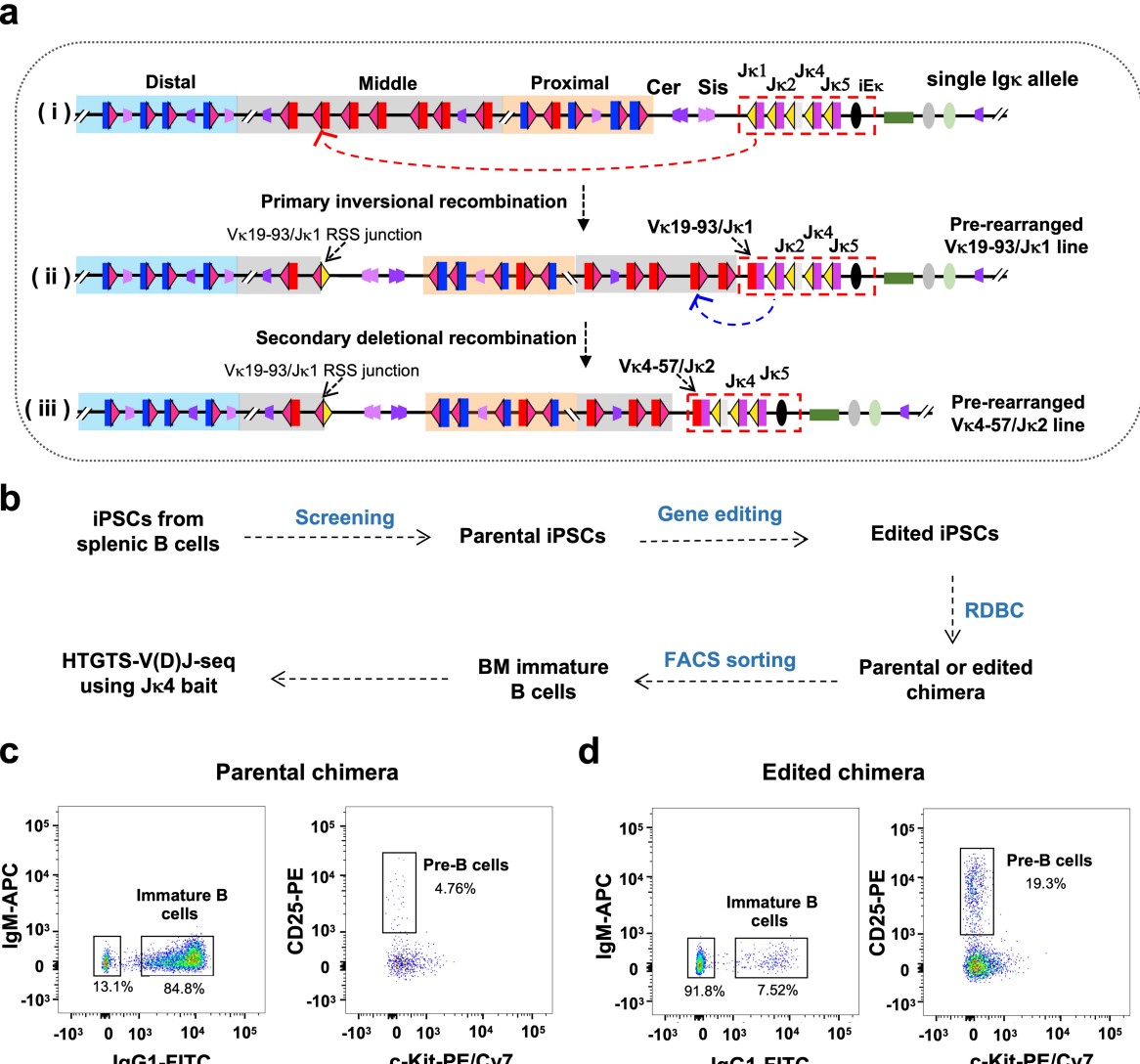

**Extended Data Fig. 5 | Generation and analysis of chimeras from splenic B cell-derived iPSCs. Related to Fig. 3. a**. Illustration of the generation of the pre-rearranged Vκ4-57/Jκ2 allele via primary inversional Vκ19-93/Jκ1 rearrangement and secondary deletional Vκ4-57/Jκ2 rearrangement. (i) Illustration of mouse *Igk*; (ii) Illustration of *Igk* pre-rearranged Vκ19-93/Jκ1 line with a primary inversional Vκ19-93/Jκ1 rearrangement that inverted the interval containing Cer/Sis, the proximal, and middle Vκ domains, placing a stretch of originally inversional-oriented middle Vκs in deletional orientation directly upstream secondary Vκ19-93/Jκ1 RC; (iii) Illustration of *Igk* pre-rearranged Vκ4-57/Jκ2 line with a secondary deletional Vκ4-57/Jκ2 rearrangement. Retained Vκ19-93/Jκ1 RSS junction and Vκ4-57/Jκ2 coding junction in the final Vκ4-57/Jκ2 iPSC line were confirmed by HTGTS-V(D)J-seq using Jκ1 RSS bait and Jκ2 coding bait respectively (Supplemental Table 2). **b**. Illustration of the strategy for generating chimeras from splenic B cell-derived iPSCs. By screening iPSCs previously generated from splenic B cells[30],

we identified a "parental" iPSC line containing a physiological productive Vκ6-23/Jκ5 rearrangement on one *Igk* allele and a nonproductive Vκ4-57/Jκ2 rearrangement on the other allele which had the potential for secondary recombination events. The productive Vκ6-23/Jκ5 rearrangement was further mutated by CRISPR/Cas9 editing to generate a second "edited" iPSC line. Both parental and edited iPSC lines were used for chimera generation via RDBC[31]. **c-d**. Flow cytometry analysis of BM B cell populations isolated from parental (**c**) and edited (**d**) chimeras. B220+CD19+IgM−IgG1− population and B220+CD19+IgM−IgG1+ population were gated and shown on the left plot. B220+CD19+IgM−IgG1− population was further used to gate the B220+CD19+ IgM−IgG1−CD25+c-Kit− population on the right plot. B220+CD19+IgM−IgG1+ population and B220+CD19+IgM−IgG1−CD25+c-Kit− population were used for defining BM immature B cells and pre-B cells respectively, as published previously (Additional details in the Methods)[19] (Supplementary Fig. 1).

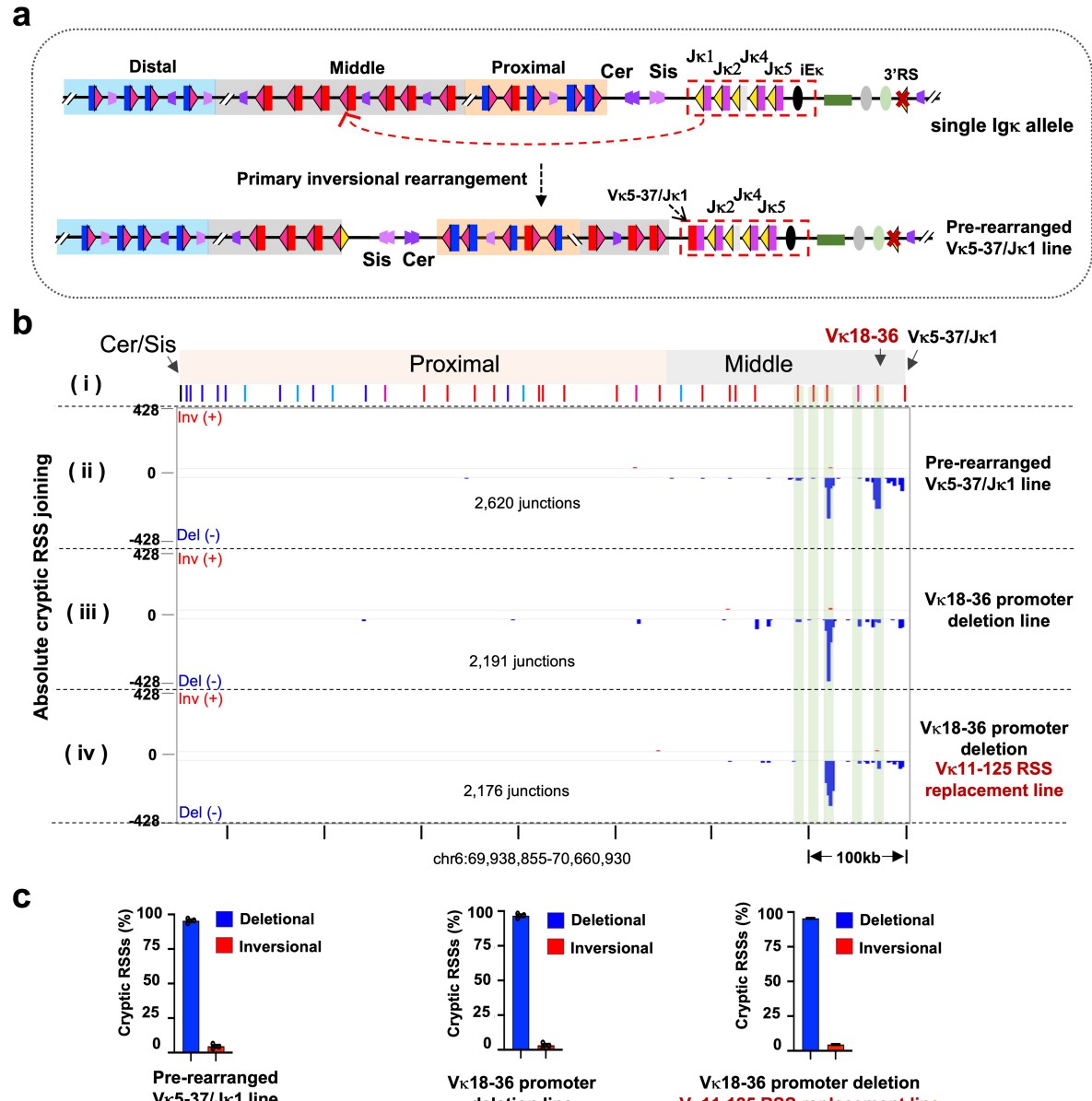

**Extended Data Fig. 6 | Analysis of cryptic RSS junctions in the pre-rearranged Vκ5-37/Jκ1 *v-Abl* cell line and its derivatives. Related to Fig. 4. a**. Illustration of the generation of the pre-rearranged Vκ5-37/Jκ1 *v-Abl* cell line via a primary inversional Vκ5-37/Jκ1 rearrangement that inverted the interval, containing Cer/Sis, the proximal, and the middle Vκ domains, placing a stretch of originally inversional-oriented middle Vκs in deletional orientation directly upstream secondary Vκ5-37/Jκ1 RC. The retained Vκ5-37/Jκ1 coding junction in the Vκ5-37/Jκ1 line was confirmed by HTGTS-V(D)J-seq using a Jκ2 coding bait (Supplemental Table 2). **b**. (i) Originally deletional (blue) and inversional (red) Vκs in the indicated region are annotated; (ii-iv) Absolute level

of pooled cryptic RSS junctions, baiting from Jκ2 in DNA from parental pre-rearranged Vκ5-37/Jκ1 line (ii), Vκ18-36 promoter deletion line (iii) and Vκ18-36 promoter deletion line with Vκ11-125 12RSS replacement (iv). Inversional cryptic joins indicated in red and deletional cryptic joins in blue. The most proximal five Vκs are highlighted (green shadow). **c**, Quantification of percentage of inversional (red) and deletional (blue) cryptic RSS junctions in the pre-rearranged Vκ5-37/Jκ1 *v-Abl* line (Left), Vκ18-36 promoter deletion line (Middle) and Vκ18-36 promoter deletion line with Vκ11-125 RSS replacement (Right). Quantification data and error bars in panel **c** are presented as mean ± s.e.m from three biological repeats.

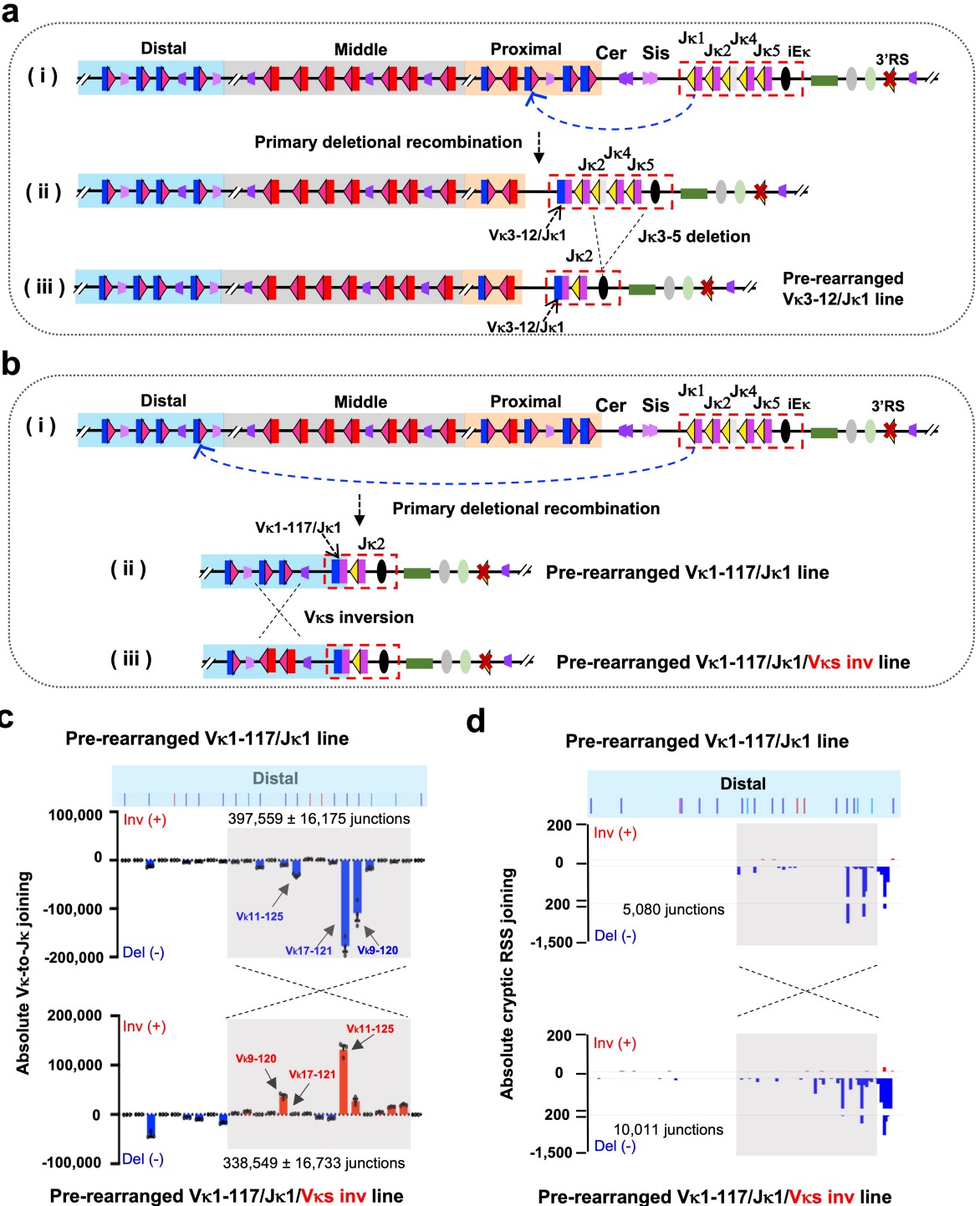

**Extended Data Fig. 7 | Robust secondary inversional Vκ-to-Jκ joining during linear RAG scanning. Related to Fig. 5. a**. (i) Illustration of mouse *Igk*; (ii) Illustration of *Igk* pre-rearranged Vκ3-12/Jκ1 *v-Abl* cell line with a primary deletional Vκ3-12/Jκ1 rearrangement that deleted the region containing Cer/Sis and a portion of proximal Vκ domain, placing inversional and deletional Vκs upstream of the secondary Vκ3-12/Jκ1 RC; (iii) Jκ3-5 were deleted by CRISPR-Cas9 editing to eliminate extra-chromosomal joining events. Retained Vκ3-12/Jκ1 coding junction in the Vκ3-12/Jκ1 line was confirmed by sanger sequencing (Supplemental Table 2). **b**. (i) Illustration of mouse *Igk*; (ii) Illustration of *Igk* pre-rearranged Vκ1-117/Jκ1 line with a primary deletional Vκ1-117/Jκ1 rearrangement that deleted the interval, containing Cer/Sis, proximal, middle and a portion of distal Vκ domains, placing distal deletional Vκs upstream the secondary Vκ1-117/Jκ1 RC; (iii) Illustration of the pre-rearranged Vκ1-117/Jκ1/ Vκs-inv line with 15 deletional Vκs upstream the secondary Vκ1-117/Jκ1 RC

inverted by CRISPR-Cas9 editing. The retained Vκ1-117/Jκ1 coding junction in the Vκ1-117/Jκ1 line was confirmed by HTGTS-V(D)J-seq using Jκ2 coding bait (Supplemental Table 2). **c**. Absolute utilization level of individual Vκs determined by HTGTS-V(D)J-seq baiting from Jκ2 in the pre-rearranged Vκ1-117/Jκ1 line (Top) or Vκ1-117/Jκ1/Vκs-inv line (Bottom). Of note, Vκ11-125 with a strong RSS (Supplementary Table 1), but not Vκ17-121 with a weak RSS (Supplementary Table 1), was robustly utilized for secondary inversional rearrangements during RAG scanning, indicating a strong RSS is required for such inversional rearrangement process. **d**. Absolute level of pooled RAG off-target junctions determined by HTGTS-V(D)J-seq baiting from Jκ2 in the pre-rearranged Vκ1-117/Jκ1 line (Top) or Vκ1-117/Jκ1/Vκs-inv line (Bottom). Inverted region is highlighted with a grey shadow. Inversional joins indicated in red and deletional joins in blue. Vκ utilization data and error bars in panel **c** are presented as mean ± s.e.m from three biological repeats.

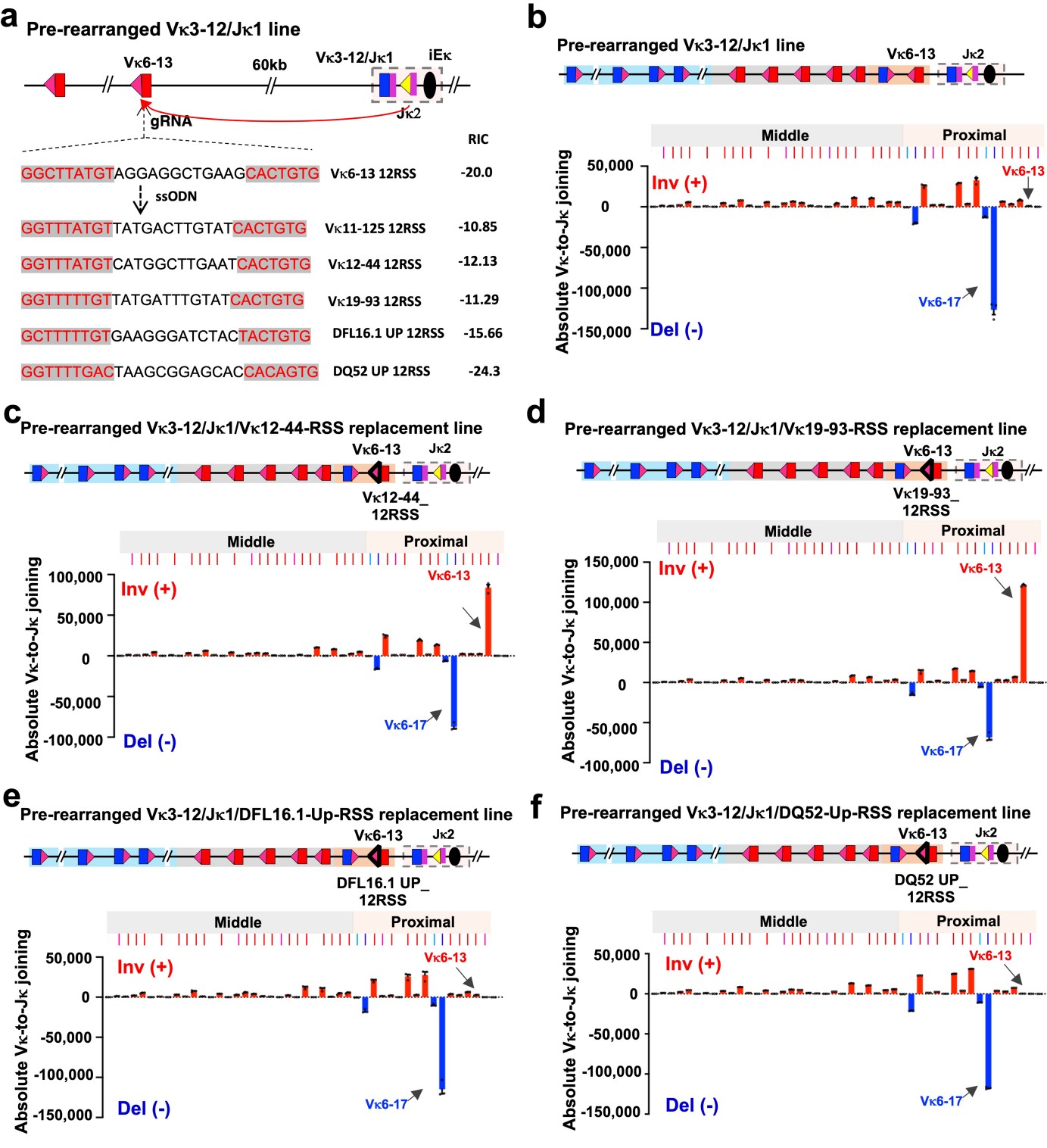

**Extended Data Fig. 8 | Strong Vκ RSSs activate secondary inversional Vκ-to-Jκ rearrangements. Related to Fig. 5.** Diagram of the strategy to replace inversional Vκ6-13 12RSS with different 12RSSs in the pre-rearranged Vκ3-12/Jκ1 *v-Abl* line. The inversional Vκ6-13 12RSS was replaced with strong or weak 12RSSs from Vκ11-125, Vκ12-44, Vκ19-93, DFL16.1-upstream and DQ52-upstream respectively, by CRISPR-Cas9-mediated homology-directed repair and confirmed by sanger sequencing (Supplemental Table 2). RIC scores of the original and the replacement RSSs are also listed on the right. **b.** Absolute utilization level of individual Vκs was determined by HTGTS-V(D)J-seq using a Jκ2 coding bait in the pre-rearranged Vκ3-12/Jκ1 line. Absolute utilization of Vκ6-17 is higher than that of the second highly utilized Vκ8-19 (two-sided Welch's t-test, P = 0.001). Absolute utilization of three inversional Vκs, Vκ8-19,

Vκ8-21 and Vκ8-24, are higher than that of the fourth highly utilized inversional Vκ6-32 (two-sided Welch's t-test, P = 0.015 for Vκ8-19, P = 0.0002 for Vκ8-21, P = 0.0046 for Vκ8-24). **c-f.** Absolute utilization level of individual Vκs was determined by HTGTS-V(D)J-seq using a Jκ2 coding bait in: (**c**) the pre-rearranged Vκ3-12/Jκ1/Vκ12-44-RSS replacement line, (**d**) the pre-rearranged Vκ3-12/Jκ1/Vκ19-93-RSS replacement line, (**e**) the pre-rearranged Vκ3-12/Jκ1/DFL16.1-Up-RSS replacement line, (**f**) the pre-rearranged Vκ3-12/Jκ1/DQ52-Up-RSS replacement line. Inversional joins are in red and deletional joins in blue. The data in panel **b** is the same as that shown in Fig. 5b (ii), plotted here for better alignment and comparison with panels **c-f** results. Vκ utilization data and error bars in panels **b-f** are presented as mean ± s.e.m from three biological repeats.

**a**

| | Jκ2 bait region | | Cryptic RSS prey region | |
|---|---|---|---|---|
| | **Jκ2 coding** | **Jκ2 23RSS** | **Cryptic RSS** | **Cryptic coding** |
| Ref sequences | 5'-CCTCCGAACGTG<u>TACA</u> | cacactggtgtcccttcactcaacccccatacaaaaact..........tccactgtg | | CC<u>ACAT</u>CACACAGAACAAAG-3' |
| **Junction numbers** | | | | |
| 32998 junctions | 5'-CCTCCGAACGTGT<u>ACA</u> | ........................................ | <u>......</u> | TCACACAGAACAAAG-3' |
| 9453 junctions | 5'-CCTCCGAACGTGT<u>A</u> | ........................................ | <u>..</u> | TCACACAGAACAAAG-3' |
| 6542 junctions | 5'-CCTCCGAACGTGTAC | ........................................ | <u>.....</u> | CACACAGAACAAAG-3' |
| 5933 junctions | 5'-CCTCCGAACGTG | ........................................ | | ACATCACACAGAACAAAG-3' |
| 5024 junctions | 5'-CCTCCGAACGTGTA<u>CA</u> | ........................................ | <u>....</u> | CATCACACAGAACAAAG-3' |
| 4889 junctions | 5'-CCTCCGAACGTGTG<u>T</u> | ........................................ | <u>..</u> | CACACAGAACAAAG-3' |

**Top 6 of Jκ2 coding-Ed cryptic coding junctions**

**b**

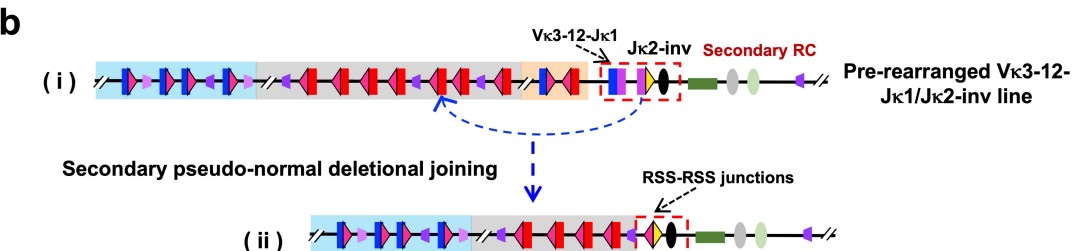

**c**

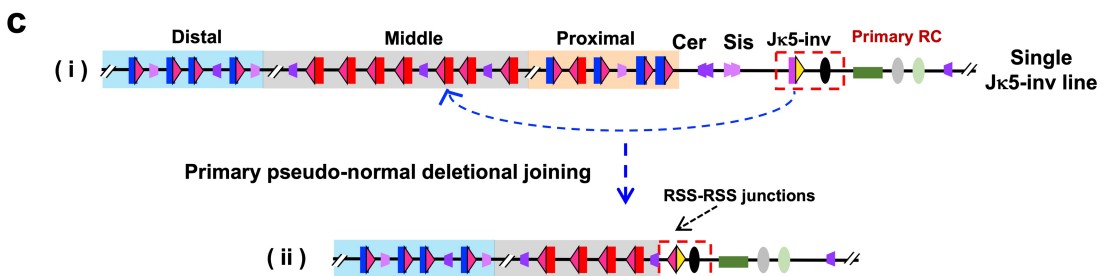

**Extended Data Fig. 9 | Illustrations of pseudo-normal deletional rearrangements during RAG scanning in the pre-rearranged Vκ3-12/Jκ1/Jκ2-inv line. Related to Fig. 5. a**. Junction sequences of a highly utilized cryptic RSS peak in the Ed enhancer region of the pre-rearranged Vκ3-12/Jκ1/Jκ2-inv line. Top 6 junctional sequences of this cryptic RSS peak mapped to either bait or prey coding reference sequences are shown. Jκ2 coding reference sequences and cryptic coding reference sequences are indicated in uppercase, and Jκ2 RSS and cryptic RSS reference sequences are indicated in lowercase. Dotted line indicates deleted sequences in each junction during *Igk* secondary rearrangements. Sequences in red indicates potential microhomologies. **b**. (i) Illustration of the *Igk* pre-rearranged Vκ3-12/Jκ1/Jκ2-inv line; (ii) Illustration of secondary "pseudo-normal" deletional joining between inversional Vκs and inverted Jκ2 in the pre-rearranged Vκ3-12/Jκ1/Jκ2-inv line. The pseudo-normal deletional V(D)J joining event excises the coding junction and retains the RSS junction on the chromosome[38,39]. **c**. (i) Illustration of the single Jκ5-inv line established previously for *Igk* primary recombination study[3]; (ii) Illustration of primary pseudo-normal deletional joining between inversional Vκs and inverted Jκ5 in the single Jκ5-inv line based on data from Fig. 3 of our prior study[3].

# Reporting Summary

## Statistics

For all statistical analyses, confirm that the following items are present in the figure legend, table legend, main text, or Methods section.

| n/a | Confirmed | |
|---|---|---|
| ☐ | ☒ | The exact sample size (*n*) for each experimental group/condition, given as a discrete number and unit of measurement |
| ☐ | ☒ | A statement on whether measurements were taken from distinct samples or whether the same sample was measured repeatedly |
| ☐ | ☒ | The statistical test(s) used AND whether they are one- or two-sided<br>*Only common tests should be described solely by name; describe more complex techniques in the Methods section.* |
| ☒ | ☐ | A description of all covariates tested |
| ☐ | ☒ | A description of any assumptions or corrections, such as tests of normality and adjustment for multiple comparisons |
| ☐ | ☒ | A full description of the statistical parameters including central tendency (e.g. means) or other basic estimates (e.g. regression coefficient) AND variation (e.g. standard deviation) or associated estimates of uncertainty (e.g. confidence intervals) |
| ☐ | ☒ | For null hypothesis testing, the test statistic (e.g. *F*, *t*, *r*) with confidence intervals, effect sizes, degrees of freedom and *P* value noted<br>*Give P values as exact values whenever suitable.* |
| ☒ | ☐ | For Bayesian analysis, information on the choice of priors and Markov chain Monte Carlo settings |
| ☒ | ☐ | For hierarchical and complex designs, identification of the appropriate level for tests and full reporting of outcomes |
| ☒ | ☐ | Estimates of effect sizes (e.g. Cohen's *d*, Pearson's *r*), indicating how they were calculated |

*Our web collection on statistics for biologists contains articles on many of the points above.*

## Software and code

Policy information about availability of computer code

Data collection | Next generation sequencing data were collected via Illumina sequencing platforms (NextSeq 550 and NextSeq 2000). NextSeq 550 control software (2.2.0) and NextSeq 1000/2000 control software (1.5.0.42699) were used for high-throughput sequencing data collection. Data generated from NextSeq 550 or NextSeq 2000 were demultiplexed via TranslocPreprocess.pl, a published pipeline available at http://robinmeyers.github.io/transloc_pipeline/.

Data analysis | HTGTS-V(D)J-seq and 3C-HTGTS data were processed via the published pipeline (http://robinmeyers.github.io/transloc_pipeline/). The pipeline for analyzing 3C-HTGTS data is available at https://github.com/Yyx2626/HTGTS_related. The pipeline for GRO-seq data analysis is available at https://github.com/Yyx2626/Fred_Alt_Lab/tree/master/GROseq. The pipeline for ATAC-seq data analysis is available at https://github.com/nf-core/atacseq. The mouse Igk-specific cryptic RSS usage analysis pipeline is available at https://github.com/Yyx2626/HTGTS_related/tree/main/Igk_specific_anno_and_filter. GraphPad Prism 9 and R 3.6.1 were used for statistical analysis and graph visualization. IGV (2.11.1) was used to visualize RAG off-target data. FlowJo (version9.3.2) was used for analyzing the FACS data.

For manuscripts utilizing custom algorithms or software that are central to the research but not yet described in published literature, software must be made available to editors and reviewers. We strongly encourage code deposition in a community repository (e.g. GitHub). See the Nature Portfolio guidelines for submitting code & software for further information.

## Data

Policy information about availability of data

All manuscripts must include a data availability statement. This statement should provide the following information, where applicable:
- Accession codes, unique identifiers, or web links for publicly available datasets
- A description of any restrictions on data availability
- For clinical datasets or third party data, please ensure that the statement adheres to our policy

HTGTS-V(D)J-Seq, 3C-HTGTS and GRO-seq sequencing data reported in this study have been deposited in the ArrayExpress database under the accession number E-MTAB-16001 for HTGTS-V(D)J-Seq data, E-MTAB-16007 for 3C-HTGTS data and E-MTAB-16014 for GRO-seq data. ATAC-seq sequencing data reported in this study have been deposited in the ArrayExpress database under the accession number E-MTAB-16602.

## Research involving human participants, their data, or biological material

Policy information about studies with human participants or human data. See also policy information about sex, gender (identity/presentation), and sexual orientation and race, ethnicity and racism.

| | |
|---|---|
| Reporting on sex and gender | N/A |
| Reporting on race, ethnicity, or other socially relevant groupings | N/A |
| Population characteristics | N/A |
| Recruitment | N/A |
| Ethics oversight | N/A |

Note that full information on the approval of the study protocol must also be provided in the manuscript.

# Field-specific reporting

Please select the one below that is the best fit for your research. If you are not sure, read the appropriate sections before making your selection.

☒ Life sciences ☐ Behavioural & social sciences ☐ Ecological, evolutionary & environmental sciences

For a reference copy of the document with all sections, see nature.com/documents/nr-reporting-summary-flat.pdf

# Life sciences study design

All studies must disclose on these points even when the disclosure is negative.

| | |
|---|---|
| Sample size | No statistical methods were used to predetermine sample size for all experiments. Sample sizes were chosen based on previous studies in this field (Dai et al., Nature 2021; Ba et al., Nature 2020; Zhang et al., Nature 2024) that used similar sample sizes to generate reproducible results. |
| Data exclusions | No data was excluded from analysis. |
| Replication | All samples were analyzed with biological three repeats as detailed in the relevant text and figure legends. All attempts for replication were successful. |
| Randomization | Experiments were not randomized. Each experiment was performed with identified control and mutant strains. Randomization was not relevant to the study as the study does not involve participant groups. |
| Blinding | Investigators were not blinded to allocation during experiments and outcome assessment. Blinding was not possible as investigators need to verify the control and matched mutant strains before each experiment. Also, based on previous studies in this field, these assays do not require blinding. |

# Reporting for specific materials, systems and methods

We require information from authors about some types of materials, experimental systems and methods used in many studies. Here, indicate whether each material, system or method listed is relevant to your study. If you are not sure if a list item applies to your research, read the appropriate section before selecting a response.

## Materials & experimental systems

| n/a | Involved in the study |
|---|---|
| ☐ | ☒ Antibodies |
| ☐ | ☒ Eukaryotic cell lines |
| ☒ | ☐ Palaeontology and archaeology |
| ☐ | ☒ Animals and other organisms |
| ☒ | ☐ Clinical data |
| ☒ | ☐ Dual use research of concern |
| ☒ | ☐ Plants |

## Methods

| n/a | Involved in the study |
|---|---|
| ☒ | ☐ ChIP-seq |
| ☐ | ☒ Flow cytometry |
| ☒ | ☐ MRI-based neuroimaging |

# Antibodies

| | |
|---|---|
| Antibodies used | anti-B220-APC (eBioscience, #17-0452-83), 1:1000<br>anti-CD43-PE (BD Biosciences, #553271), 1:400<br>anti-IgM-FITC (eBioscience, #11-5790-81), 1:500<br>anti-B220-BV711 (BioLegend, Cat#103255), 1:300<br>anti-CD25-PE (BD PharmingenTM, Cat#561065), 1:300<br>anti-IgG1-FITC (BD Biosciences, Cat#553443), 1:500<br>anti-IgM-APC (Invitrogen, Cat#17-5790-82),1:500<br>anti-CD19-BV421 (BD Biosciences, Cat#562701), 1:300<br>anti-c-Kit-PE/Cy7 (eBioscienceTM, Cat#25-1171-81) , 1:300 |
| Validation | anti-B220-APC (eBioscience, #17-0452-83), anti-CD43-PE (BD Biosciences, #553271) and anti-IgM-FITC (eBioscience, #11-5790-81) have been confirmed by FACS in published papers including (except this study): Dai, H.-Q. et al. Loop extrusion mediates physiological Igh locus contraction for RAG scanning. Nature 590, 338–343 (2021).<br><br>anti-B220-BV711 (BioLegend, Cat#103255), anti-CD25-PE (BD PharmingenTM, Cat#561065), anti-IgG1-FITC (BD Biosciences, Cat#553443), anti-IgM-APC (Invitrogen, Cat#17-5790-82), anti-CD19-BV421 (BD Biosciences, Cat#562701) and anti-c-Kit-PE/Cy7 (eBioscienceTM, Cat#25-1171-81) have been confirmed by FACS in published papers including(except this study): Hill, L. et al. Wapl repression by Pax5 promotes V gene recombination by Igh loop extrusion. Nature 584, 142-147 (2020). |

# Eukaryotic cell lines

Policy information about cell lines and Sex and Gender in Research

| | |
|---|---|
| Cell line source(s) | The primary pre-B cells were derived from bone marrows of 4-8-week-old WT mice, Cer/Sis-deleted mice and iPSC-derived chimeras in both sex. The iPSC cell lines and derivatives were generated by reprogramming from splenic B cells, made in our lab. All immortalized v-Abl cell lines and derivatives were generated by retroviral infection of primary pro-B cells derived from RAG2-deficient; Em-Bcl2 transgenic male 129SV mice with pMSCV-v-Abl retrovirus, made in our lab. See Methods for details. |
| Authentication | All cell lines were authenticated by HTGTS-V(D)J-Seq using indicated baits, PCR genotyping and Sanger sequencing. See Methods for details. Sequences of pre-rearranged Vk/Jk segments and targeted genome modifications are listed in Supplementary Table 2. Sequences of all sgRNAs and oligos used are listed in Supplementary Table 3. |
| Mycoplasma contamination | All iPSC lines used for targeting and RAG-deficient blastocyst complementation injections were confirmed to be mycoplasma free. v-Abl cell lines were not tested for mycoplasma contamination. |
| Commonly misidentified lines<br>(See ICLAC register) | None |

# Animals and other research organisms

Policy information about studies involving animals; ARRIVE guidelines recommended for reporting animal research, and Sex and Gender in Research

| | |
|---|---|
| Laboratory animals | We used 4-8-week-old WT mice and Cer/Sis-deleted 129SV mice, including both males and females, for isolating primary pre-B cells from bone marrow. We used 4-8-week-old iPSC-derived chimeras containing mixed genetic background of C57BL/6 and 129SV, including both males and females, for isolating primary immature B cells from bone marrow. |
| Wild animals | The study did not involve wild animals. |
| Reporting on sex | Both male and female mice were used in experiments. |
| Field-collected samples | The study did not involve samples collected from the field. |
| Ethics oversight | All mouse work were performed in compliance with all the relevant ethical regulations established by the Institutional Animal Care and Use Committee (IACUC) of Boston Children's Hospital and under protocols approved by the IACUC of Boston Children's Hospital. |

Note that full information on the approval of the study protocol must also be provided in the manuscript.

# Plants

Seed stocks | N/A

Novel plant genotypes | N/A

Authentication | N/A

# Flow Cytometry

## Plots

Confirm that:

☒ The axis labels state the marker and fluorochrome used (e.g. CD4-FITC).

☒ The axis scales are clearly visible. Include numbers along axes only for bottom left plot of group (a 'group' is an analysis of identical markers).

☒ All plots are contour plots with outliers or pseudocolor plots.

☒ A numerical value for number of cells or percentage (with statistics) is provided.

## Methodology

Sample preparation | Single cell suspensions were derived from bone marrows of 4-8-week-old iPSC-derived chimeras, incubated in Red Blood Cell Lysing Buffer (Sigma-Aldrich, #R7757) to deplete the erythrocytes. Immature B cells were isolated by staining with anti-B220-BV711 (BioLegend, Cat#103255), anti-CD25-PE (BD PharmingenTM, Cat#561065), anti-IgG1-FITC (BD Biosciences, Cat#553443), anti-IgM-APC (Invitrogen, Cat#17-5790-82), anti-CD19-BV421 (BD Biosciences, Cat#562701) and anti-c-Kit-PE/Cy7 (eBioscienceTM, Cat#25-1171-81) antibodies for 30 minutes at 4 °C and then purified by FACS.

Instrument | BD FACSAria II

Software | FlowJo vX.0.7

Cell population abundance | The cell populations of Pre-B cells and Immature B cells in the BM marrow of iPSC-derived chimeras are indicated in Extended Data Fig.5 c, d.

Gating strategy | Lymphocyte population was gated by FACS side (SSC) and forward (FSC) scatters out of the total cells. Then B220+CD19+ B cells were gated and selected by FACS. IgG1+IgM- immature B cells were gated and shown in the left plot (Extended Data Fig.5 c, d). IgG1-IgM- B cells were also gated, and used for gating CD25+c-Kit- pre-B cells indicated in the right plot (Extended Data Fig.5 c,d).

☒ Tick this box to confirm that a figure exemplifying the gating strategy is provided in the Supplementary Information.

