## [Peer Review File · Nature]

Linear RAG Scanning Mediates Editing of Igk Variable Region Repertoires

Corresponding Author: Professor Frederick Alt

Version 1:

Reviewer comments:

Referee #1

(Remarks to the Author)

The manuscript by Li et al. is a well-executed study that offers a clear mechanistic explanation of secondary V(D)J recombination at the immunoglobulin kappa (Igk) locus. In recent years, loop extrusion has been demonstrated to be the fundamental mechanism of immune receptor diversification. The kappa locus, which contains V gene cassettes with both orientations, poses a big puzzle to our understanding of the molecular mechanisms. This study fills this last gap, providing a detailed depiction of receptor editing mechanism that previous works never achieved.

In this study, the authors first delete the Cer/Sis elements to mimic a primary rearrangement in mouse pre-B cells, then examine RAG on- and off-target junctions using HTGTS-V(D)J-seq. By measuring the relatively low level of joins between Jk RSSs and upstream cryptic RSSs, they found that RAG scanning generates limited utilization of upstream Vks.

Furthermore, the same phenotype was observed using the new Vk4-57/Jk2 receptor editing mouse models. Mechanistically, the authors used modified v-Abl cell lines to validate the roles of transcription and RSS strength in this process. Collectively, these findings provide compelling evidence that secondary Igk recombination follows a linear scanning-based mechanism. Overall, the study elegantly illustrates how the Igk locus has evolved distinct mechanistic solutions for primary (diversity-focused) and secondary (editing-focused) recombination processes.

Specific comments and suggestions:

1. The Panel a-d in Fig. 1 and Fig. 2 are the same, which is convenient for the reviewers to understand the data, as the authors kindly noted in the figure legends. However, the same panels should be deleted in the revision. In addition, Fig. 2f should be better labelled.
2. The authors proposed a convincing model explaining the secondary Jk2-5 recombination. This is an important point, and an illustration could be drawn in Fig. 2e to explain the data. Whether remnant junctions (158) between Jk2 and cryptic RSSs (Sis-Jk) are resulted from primary recombination?
3. Figures should be better labeled. In Fig. 4/5, Panel a, (i)-(iv) nicely indicate different genetic modified lines. In Panel c/d, the labeling should be kept the same. In the Fig 3, Jk elements should be better illustrated, e.g. using lines to indicated the elements.
4. In Line 228-231/249-250, the authors should better discuss the differences between IgH and IgK loci. For example, whether CBE elements are presented at IgK at a similar frequency. In addition, the authors are welcome to discuss the evolution of immune receptor loci in mammals.
5. Overall, the authors proposed a novel model of receptor editing, in which only a few immediate upstream Vks can be used for secondary recombination. Whether this mechanism plays a physiological role? For example, whether similar Vks are clustered and the receptor editing likely produces a successful H-L pair. The authors could discuss different hypotheses, although the validation is out of the scope of current manuscript.
6. There are some typos. For example, 3'RSS should be 3'RS in Line 183/240.

Referee #2

(Remarks to the Author)

Linear RAG Scanning Mediates Editing of Igk Variable Region Repertoires

It is now well established that Igh locus V(D)J recombination occurs by deletion, whereas Igk locus rearrangement involves either deletion or inversion. Igh locus VHDHJH rearrangement is mediated predominantly by RAG scanning whereas primary VkJk recombination is instructed by diffusion that Vk and Jk elements that are positioned into two distinct loop configurations. How secondary Vk-Jk rearrangements to remove self-reactive or non-functional VkJk rearrangements are orchestrated during the course of receptor revision has remained to be determined.

In this manuscript the authors show that deletional and inversional VkJk1 primary rearrangements, delete or displace a critical cis-regulatory elements, to generate pre-B cells that leaves the majority of Vk gene segments positioned in a deletional configuration upstream of the de novo (secondary) recombination center. The authors show that this remaining set of Vk genes undergo RAG scanning to generate a diverse set of VkJk rearrangements. In a parallel study the authors validate these findings by utilizing iPSC mouse models as well as Abelson lines that carry VkJk rearrangements to further boost support for a RAG scanning mechanism in the receptor revision process. Collectively, these data provide compelling evidence the receptor revision process involves a RAG-scanning mechanism.

How during the course of B cell progression, a tolerant and diverse immunoglobulin repertoire is assembled at the mechanistic level has remained an open question. In this study new and important insights are provided. The data are robust and compelling. The manuscript is also well written and results are presented in a well-organized fashion. I only have a few comments that could further improve the manuscript.

Major comments:

Could the authors please include HiC maps to confirm expected differences in Vk-Jk chromosomal contacts for wild-type and Cer-Sis deleted pre-B cells?

It would be of interest to determine whether and how chromatin accessibility is altered across the recombination signal sequences in wild-type and Cis-Sis deleted pre-B cells.

Minor comment:

References should perhaps be a bit more balanced and include those of others that also have contributed to this field.

Referee #3

(Remarks to the Author)

The manuscript by Li et al presents a convincing mechanism by which gene segments are brought into close proximity for secondary recombination in the mouse immunoglobulin kappa locus (Igk). The Igk locus is unusual in that it undergoes both deletional and inversional recombination. The authors previously presented a two-loop extrusion model for primary Igk recombination whereby cohesion-mediated loop extrusion allows RAGs bound to Jk1 to linearly scan upstream Vk gene segments but only as far as the Cer/Sis element. Recombination to more distal Vk RSSs required requires a second loop through which Vk RSSs can encounter the Jk1-bound RAGs via short-range diffusion. Here, the authors examined how secondary recombination occurs to Jk2, Jk4 and Jk5 using high throughput sequencing and gene editing approaches. They convincingly show that following primary recombination events that either delete the Cer/Sis element or recombine it such that it is distal to the originally inversional middle Vk RSSs, RAGs are able to linearly scan upstream Vk RSSs in a one-loop extrusion model, including middle RSSs that now undergo deletional recombination. There are some exceptions where RSSs in the inversional orientation lie next to the recombination centre and in these cases, the authors propose a short-range diffusion model.

The experiments are carefully performed and well presented. Given the high frequency of secondary recombination, the data explain an important and fundamental step in the generation of the antigen receptor repertoire. As such, they are likely to be of broad interest to people in the V(D)J recombination field and have implications for long range chromatin interactions that will be of interest to colleagues studying higher order chromatin folding/interactions. The outstanding features are the neat use of iPSC and well controlled gene editing experiments.

Although this is an extremely interesting and well presented manuscript, there are some points that should be addressed.

Major points:

- 1) The authors neatly show how secondary recombination to Jk2, Jk4, and Jk5 can occur. However, there are also substantial amounts of secondary rearrangement to the RS element (Yamagami et al., 1999). This should be discussed.
- 2) To highlight the importance of secondary recombination to the Igk repertoire, it would be very helpful if the authors cite the levels of secondary recombination that have been determined such as in the single cell analyses by Yamagami et al (1999).
- 3) Could the authors please clarify why, in Fig 1g, scanning from Jk5 only extends for a limited distance into the middle Vks upon deletion of the Cer/Sis element whereas in Fig 2f, scanning from Jk5 extends across the locus?
- 4) Figure 4 and lines 206-207. The authors claim that transcription is an impediment to linear scanning. However, it is the highly transcribed Vks that undergo most rearrangement (Vk6-32, Vk8-34 and Vk18-36) and removal of the Vk18-36 promoter substantially reduces its recombination. Could this also suggest that transcription is NEEDED for recombination?

What is the evidence that transcription is an impediment to linear scanning? This must be addressed.

5) Extended Data Figure 4c: Can the authors explain why recombination to Vk17-121 is almost completely lost in the inverted orientation? Conversely, why does recombination to Vk11-125 show a greater increase relative to the other Vks analysed in the inverted orientation?

6) It would be helpful to include the significance of various changes in recombination levels in the figure legends as this group has done in previous publications. For example:

- Line 170: “edited chimeras were highly enriched for productive Vks/Jk4 rearrangements” Figs 3c,f. The significance of the increase in productive rearrangements would be useful.

- Line 209: “which markedly reduced Vk18-36 rearrangement” Fig 4biii. The significance of the decrease would be useful.
- Line 242: “deletional-oriented Vk6-17 is the most robustly rearranged; but, at least three inversional Vks, including Vk8-24 about 300kb upstream, are significantly utilized” Please give the significance and what it is being compared to.

Minor points:

1) Fig 5a labelling: “inversion” should read “inversion”

2) It would be helpful to refer to the RS element as such, rather than 3' RSS. As written, this is confusing.

3) Line 121: The authors claim RAG scanning from the primary Jk1-RC is terminated at Sis. However, Fig. 1c reports 219 junctions beyond Sis. Could the authors please add more explanation, perhaps based on the levels of scanning beyond Sis. It is hard to compare with the levels in Fig. 2f as the scales on the y axes are different.

4) Supplementary Table 2 contains the coding sequences of the V-J joins but it is not clear which sides of the sequence as well as the breakpoint which makes it hard to interpret. These could be highlighted to allow for better understanding. Further, there appears to be no sequence tracks for any of the following derivative lines:

- the pre-rearranged Vk5-37/Jk1 v-Abl cell line to generate its Vk18-36 promoter-deleted derivative;
- the pre-rearranged Vk1-117/Jk1 v-Abl cell line to generate a Vks-inverted derivative
- the pre-rearranged Vk3-12/Jk1 v-Abl cell line to delete the DNA segment from upstream of Jk3 through downstream of Jk5;
- the pre-rearranged Vk3-12/Jk1 v-Abl cell line to generate its Jk2-inverted derivative. All candidate clones with desired genetic modifications were confirmed by Sanger sequencing (line 467)

Comments on Code:

I am not an expert to review the code but all code seems accessible from the links provided. However, the method of Gro-Seq analysis was not given in the Code Availability statement. Please can this be added?

Referee #4

(Remarks to the Author)

I co-reviewed this manuscript with one of the reviewers who provided the listed reports.

Version 2:

Reviewer comments:

Referee #1

(Remarks to the Author)

The authors satisfactorily addressed all of my previous comments.

Referee #2

(Remarks to the Author)

Authors addressed critiques and comments raised by reviewers resulting in a much improved manuscript.

Referee #3

(Remarks to the Author)

The authors have fully addressed my previous concerns. I thank them for doing such a thorough job.

Just in case the per review files are released, it should be noted that the response to Reviewer 3, comment 5 states that the modified text is given in the legend to Extended Data Fig 6c (lines 959-963). It is in fact given in the Legend to Extended data 7c, lines 942-945. The manuscript is fine - a minor edit to the response letter would help if it is to be released.

Referee #4

(Remarks to the Author)

I co-reviewed this manuscript with one of the reviewers who provided the listed reports.

Response to Editor

Dear Dr Alt,

Your manuscript, "Linear RAG Scanning Mediates Editing of Igk Variable Region Repertoires", has now been seen by three principal reviewers and one co-reviewer, whose comments are attached below. While they find your work of potential interest, as do we, they have raised a number of points that we should like to see addressed before we can consider publication in Nature.

We hope to receive your revised paper within four to six months. If you cannot complete the required revisions within this time frame, please let us know when you would anticipate being able to submit a revised manuscript.

Any revised manuscript should conform to our format instructions and publication policies (see here). We also strongly suggest that your revised manuscript has tracked changes, which is increasingly requested by referees to aid in their re-review.

We have highlighted changes in Red in the revised manuscript.

We would appreciate your careful attention to the following:

STATISTICS: When revising your manuscript, you should ensure that any statistical analysis used is sound and that it conforms to Nature's guidelines). A collection of articles explaining the basics of statistical analysis and advice on how to best present it can be found here.

We have confirmed that our statistical analyses conform to Nature guidelines.

REPRODUCIBILITY: All of the checklists provided with the current submission (Reporting summary, and Code and software checklist (if applicable)) should be updated to reflect the revisions made and submitted with the revised manuscript.

We have updated the checklists to reflect revisions made in the revised manuscript.

DATA AND CODE AVAILABILITY STATEMENTS: All original research manuscripts published in Nature Portfolio journals must include a Data availability statement. This statement must make the conditions of access to the "minimum dataset" that is necessary to interpret, verify and extend the research in the article, transparent to readers. This minimum dataset may be provided through deposition in public community/discipline-specific repositories, custom proprietary repositories for certain types of datasets, or general repositories like Figshare, Zenodo and Dryad. Providing large datasets in Supplementary Information is strongly discouraged; the preferred approach is to make data available in repositories. More information on Nature Portfolio's reporting standards and preparing your Data availability statement can be found here.

For all studies using custom code or mathematical algorithms that are deemed central to the conclusions, a Code availability statement must be included, indicating whether and

how the code or algorithm can be accessed, including any restrictions to access. The Code availability statement should be provided as a separate section after the Data availability statement but before the references. Code should be deposited in a DOI-minting repository such as Zenodo, Gigantum or Code Ocean and cited in the reference list. We encourage you to manage subsequent code versions and to use a license approved by the open source initiative. Additional details can be found here.

We have provided a Data Availability statement and a Code Availability statement in the Methods section. We have deposited the sequencing data in the ArrayExpress database and have deposited data analysis pipelines in GitHub.

METHODS: After the main text figure legends there should be a section entitled "Methods", which provides the full, step-by-step instructions that would allow other researchers to replicate the results. The Methods section will not appear in print but will appear online in the full-text HTML and PDF versions. The Methods section should be written as concisely as possible but should contain all elements necessary to allow interpretation and reproduction of the results. If there are additional references in the Methods section, their numbering should continue from the last reference in the main text, and they should be listed following the Methods section. Specialized methods that require chemical structures, figures or tables, or methods requiring equations, cannot be accommodated in the Methods section of the main text file. If such information is part of the Methods, the entire Methods section must instead be included within a Supplementary Information text file.

We confirm that the Methods section is written as concisely as possible but contains all elements necessary to allow interpretation and reproduction of the results. Additional References in the Methods section are listed following the Methods section, with continued numbering from the last reference in the main text. We do not have specialized methods in this manuscript.

EXTENDED DATA: Extended Data do not appear in the print version of the paper but are included online within the full-text HTML and at the end of the online PDF. Extended Data are an integral part of the paper, and only data that directly contribute to the main message should be included. All Extended Data must be referred to in the main text, and their legends should be listed sequentially at the end of the main text, not in the Extended Data files. Extended Data should be assembled into a maximum of 10 A4 size, multi-panelled display items, submitted as individual files in .jpg, .tif or .eps format only. They should be of the same quality as print figures, but there are important differences in their formatting. More specific instructions are provided here. If you need to describe complex processes, we encourage you to include a schematic of the main finding as part of the Extended Data to aid readers unfamiliar with the immediate discipline.

We currently have 9 Extended Data figures with legends listed sequentially at the end of the main text. All Extended Data figures are referred to sequentially in the main text. We confirm that all Extended Data figures conform to Nature's guidelines. We will submit individual Extended Data figures in .jpg format. We have included schematics of our main findings as part of the Extended Data to aid readers unfamiliar with the immediate discipline.

SUPPLEMENTARY INFORMATION: Supplementary Information (SI) is online-only, peer-reviewed material that is essential background to the study (e.g., large data sets, more complex methods, and calculations), but which is too large or impractical, or of interest only to a few specialists, to justify inclusion in the print version of the paper (see here for further details). While SI should not typically contain data figures (any figures additional to those appearing in the main text should be formatted as Extended Data), we require that the raw, uncropped data for gels be presented as an SI figure (see below). Tables may be included in SI, but only if they are unsuitable for formatting as Extended Data (e.g., tables containing large data sets or raw data tables that are best suited to Excel files). If a manuscript has SI, each discrete item of the SI (e.g., videos, tables) must be referred to at an appropriate point in the main manuscript.

We currently have three Supplementary Tables and four Supplementary Videos. All of them are referred to in the main text or Methods section. We have provided an "SI Guide" word file for navigation.

SOURCE DATA (GRAPHS): To increase transparency, we ask you to provide, in spreadsheet form, the data underlying the graphical representations used in figures. In the case of all experiments presenting data from animal models, this is a requirement and is not optional. This is in addition to our well-established data-deposition policy for specific types of experiments and large datasets. Online readers of the manuscript will be able to access the graphical source data directly from the figure legend. Spreadsheets must be submitted in .xls, .xlsx or .csv formats. One file per figure is permitted. If there is a multi-panelled figure, the source data for each panel should be clearly labeled in the file; alternatively the source data for a figure can be included in multiple, clearly labeled sheets within an Excel file. File sizes of up to 30 MB are permitted, but it is expected that the vast majority of graphical source data files will be considerably smaller than this. When submitting these files with your manuscript, you should select the "Source Data" file type and use the title field in the file description tab to indicate the figure(s) to which the source data pertain.

We have provided source data files for all main text and Extended Data figures in Excel format. All source data files are within 30 MB.

RAW DATA (GELS): You must provide the original source images for all data obtained by electrophoretic separation (e.g., EMSA, northern/Southern/western blots, etc). The raw images should be assembled into a single .pdf or .tif file (multiple gels on a single page is encouraged). The file should be uploaded as Supplementary Figure 1. The full scanned images must be in uncropped form and contain labeled size/molecular weight markers and loading controls. There should be an accurate indication of how the gels were cropped for the final figure. The figure legends and raw data files should indicate whether controls (such as beta-actin) were run on the same gel as loading controls or on separate gels as sample processing controls (see here). While the data can be displayed in a relatively informal style, there must be a correspondence between each source data image and a specific main text or Extended Data figure. The main text or Extended Data figure legends should refer to the uncropped scans explicitly (e.g., "For gel source data, see Supplementary Figure 1."). For examples, see here or here.

We do not present any data obtained by electrophoretic separation in this manuscript.

THIRD PARTY RIGHTS: Please identify any content used in your article, whether in the main text, Extended Data or Supplementary Information, that comes from a third party. This could include figures, tables, images, videos or text boxes that are reproductions or adaptations of items that have previously been published elsewhere and/or are owned by a third party. It also encompasses pictures taken by professional photographers, maps and images downloaded from the internet. You must obtain the right to use each of these items and provide evidence that you have these rights. You will also need to give proper attribution to the copyright holders in your paper. We ask that you fill out a Third party rights table and upload this with your revised manuscript. You can find more information about third-party rights here.

We have included a Third party rights table for Supplementary Video 1. Supplementary Video 1 has been published before (Yu Zhang et al., Nature, 2019) from my lab to diagram the model for IgH V(D)J recombination and put here again to better compare with three new Supplementary Videos that diagram the models for Ig κ primary diffusional and secondary scanning rearrangements.

Referees' comments:

Referee #1 (Remarks to the Author):

The manuscript by Li et al. is a well-executed study that offers a clear mechanistic explanation of secondary V(D)J recombination at the immunoglobulin kappa (Igk) locus. In recent years, loop extrusion has been demonstrated to be the fundamental mechanism of immune receptor diversification. The kappa locus, which contains V gene cassettes with both orientations, poses a big puzzle to our understanding of the molecular mechanisms. This study fills this last gap, providing a detailed depiction of receptor editing mechanism that previous works never achieved.

In this study, the authors first delete the Cer/Sis elements to mimic a primary rearrangement in mouse pre-B cells, then examine RAG on- and off-target junctions using HTGTS-V(D)J-seq. By measuring the relatively low level of joins between Jk RSSs and upstream cryptic RSSs, they found that RAG scanning generates limited utilization of upstream Vks. Furthermore, the same phenotype was observed using the new Vk4-57/Jk2 receptor editing mouse models. Mechanistically, the authors used modified v-Abl cell lines to validate the roles of transcription and RSS strength in this process. Collectively, these findings provide compelling evidence that secondary Igk recombination follows a linear scanning-based mechanism. Overall, the study elegantly illustrates how the Igk locus has evolved distinct mechanistic solutions for primary (diversity-focused) and secondary (editing-focused) recombination processes.

Response:

We thank this referee for the very careful review of our manuscript. We greatly appreciate the referee's highly favorable evaluation of the quality and significance of our work.

Specific comments and suggestions:

1. The Panel a-d in Fig. 1 and Fig. 2 are the same, which is convenient for the reviewers to understand the data, as the authors kindly noted in the figure legends. However, the same panels should be deleted in the revision. In addition, Fig. 2f should be better labelled.

Response:

Thanks very much for pointing this out. In the originally submitted manuscript, we noted this duplication in the Figure legends for reasons mentioned by the referee. However, we also well-understand that some who view the manuscript may not read all the details in the Figure Legends. So, as suggested by this referee, we have removed panels a-d in Figure 2 and now refer to these panels in Figure 1 in the text to make the comparison.

2.a. The authors proposed a convincing model explaining the secondary Jk2-5 recombination. This is an important point, and an illustration could be drawn in Fig.2e to explain the data.

Response:

Thanks for pointing this out. We agree that this is a very important point. We had illustrated this point previously in Extended Data Figure 1 (now revised Extended Data Figure 4). We now have further clarified this point in the text describing Figure 2 (line 154-156) as follows: "*These findings indicate that limited linear scanning from the numerous secondary V κ J κ 1-based RCs established during primary joining collectively covers the entire locus. (Extended Data Figure 4 and legend)*". We also further clarified this point in the revised Extended Data Figure 4 Legend.

2b. Whether remnant junctions (158) between Jk2 and cryptic RSSs (Sis-Jk) are resulted from primary recombination?

Response:

Good point! Yes, the data in the previous Fig.2f (now Fig.2e in the revised manuscript) suggests the possibility of a very low level of primary rearrangements from J κ 2. We have now made this point in the text (line 138-139): "*However, we did observe a low level of impeded scanning activity from J κ 2 to Sis (Fig.2e,h), suggesting that it may also weakly contribute to primary rearrangements*". We note that the number of remnant junctions in this figure is now listed as 137. When examining very low-level RAG scanning (cryptic RSS) patterns for a different study in progress, we found that a minor fraction of these low-level junctions represented artifacts associated with mapping surrogate coding sequence to J κ coding sequence joins. Based on that, we very recently redesigned a refined Ig κ -specific cryptic RSS pipeline to eliminate such low-level artifacts. We now have applied this refined pipeline to all of our figures in the current study that report RAG scanning data and profiles. While eliminating a very low-level artifactual junctions, the application of the refined pipeline had very little, if any, effect on overall cryptic RSS patterns and did not affect any interpretations or conclusions. As we had not yet published this refined Ig κ pipeline, we have described it in the revised methods of the current study and made its code available, which includes annotation of the data on which revisions were based.

3. Figures should be better labeled. In Fig. 4/5, Panel a, (i)-(iv) nicely indicate different genetic modified lines. In Panel c/d, the labeling should be kept the same. In the Fig 3, Jk elements should be better illustrated, e.g. using lines to indicated the elements.

Response:

Thanks for pointing this out. We have addressed all of these points in the revised figures.

4.a. In Line 228-231/249-250, the authors should better discuss the differences between IgH and IgK loci. For example, whether CBE elements are presented at IgK at a similar frequency.

Response:

We thank the referee for raising this point. To address this comment, we have now added relevant sentences to the main text (line 252-255) stating: "While we cannot rule out a modest role for V_K locus CBEs, we note that V_K locus CBEs are less dense and less potent than V_H locus CBEs³, particularly those of proximal V_H s that play a direct role in linear scanning¹⁶ (See also Extended Data Fig.1 Legend)". In addition, we have now included a new Extended Data Fig.1 to diagram different V(D)J recombination models in the IgH and IgK locus. We have discussed all of these points in detail in the Extended Data Fig.1 legend (line 762-813) and provided additional references as follows:

“Extended Data Fig.1 One-loop scanning-based versus two-loop diffusion-based mechanisms for IgH and IgK V(D)J recombination.

(Related to Figs.1-5). a. One-loop-based linear RAG scanning model for V_H -to- DJ_H rearrangements.

Active chromatin of the transcribed nascent Igh RC serves as a dynamic barrier to impede downstream cohesin-mediated loop extrusion and recruit RAG, which binds a J_H RSS to form an active IgH RC^{14,16}. This one-loop J_H -based IgH RC programs RAG to linearly scan the upstream D-containing 100kb domain from the RC to the IGCR-1 CBE-based loop anchor, only using the downstream D-12RSSs in convergent orientation with the J_H -23RSS, despite compatibility of upstream D-12RSSs for joining^{14,17} (Supplemental Video 1). Joining establishes a new D- J_H RC¹⁶. Upon WAPL down-regulation, which partially neutralizes IGCR1 CBE-based scanning impediments^{19,20,43}, scanning moves upstream chromatin from the DJ_H RC to a block of proximal V_H s with strong CBEs that lie within 20bp of their RSSs¹⁶. Due to weak Igh RSSs, proximal V_H s require RSS-associated CBEs to promote interactions with the RC for rearrangement¹⁶. As

Extended Data Fig.1 (Related to Fig.1-5)

WAPL down-regulation partially neutralizes proximal V_H -RSS CBEs, scanning proceeds into the distal $V_{HS}^{20,43}$. Distal V_{HS} lack closely associated CBEs but are mostly transcribed^{16,18,20}. Transcription, a well-known dynamic loop extrusion impediment^{17,44-46}, facilitates their interaction with the RC and their rearrangement¹⁷. During one-loop linear RAG scanning, only Bonafide D or V_H RSSs in convergent orientation are robustly used due to the weakness of Igh RSSs^{3,17,20}. **b. Two-loop-based diffusional model for primary $V\kappa$ -to- $J\kappa I$ rearrangements.** *Cer/Sis* CBEs are more potent than those of *IGCR1*, which generates a platform for diffusional access of $V\kappa$ s as they are moved past the closely associated $J\kappa I$ -based RAG-bound primary RC³. Such long-range extrusion of the $V\kappa$ locus can happen even with very high WAPL-levels in *v-Abl* cells, because CBEs in the $V\kappa$ locus are less dense and less potent than those in those in V_H locus³. For primary diffusion-based $V\kappa$ -to- $J\kappa I$ rearrangements, CBEs and transcription, and other scanning impediments may enhance transient interactions with strong *Cer* CBEs that help promote transient diffusion-based access of $V\kappa$ s to the primary $J\kappa I$ -RC³. For this process, very strong $V\kappa$ and $J\kappa$ RSSs are required to promote joining³. **c. One-loop-based linear RAG scanning model for secondary deletional and (previously) inversional $V\kappa$ joining.** Primary $V\kappa$ to $J\kappa I$ rearrangements delete or displace *Cer/Sis* creating $V\kappa J\kappa I$ -based secondary RCs at every rearranged $V\kappa$ across the locus in a population of pre-B cells. Most deletional and inversional $V\kappa$ to $J\kappa I$ joins also leave blocks of deletional- or previously inversional-oriented $V\kappa$ s, in deletional orientation upstream of secondary $V\kappa J\kappa I$ -RCs that employ $J\kappa S2,4$, and $5^{4,5,10}$. Because of *Cer-Sis* deletion/displacement, the $Ig\kappa$ secondary $V\kappa J\kappa I$ -RC serves as a dynamic loop anchor that employs direct, one-loop based, linear RAG scanning similar to that used by *Igh* for $V(D)J$ recombination. Notably, besides having less potent CBEs than *Igh*, none of the $V\kappa$ s have RSS-associated CBEs that would promote their direct RC interaction^{3,47}. Finally, direct scanning-based $Ig\kappa$ secondary rearrangements, like $Ig\kappa$ primary rearrangements, occur in the presence of robust WAPL levels that block nearly all V_H rearrangements in the same cells^{3,20}. While there is no evidence that rules out a role for CBEs in promoting $Ig\kappa$ secondary rearrangements, there is no evidence that directly supports such a role. However, our current findings implicate $V\kappa$ transcription in increasing association with the RC¹⁷ and also in mediating its chromatin accessibility²⁵⁻²⁹, both of which, coupled with strong $V\kappa$ RSSs³, likely lead to saturation of secondary rearrangements in a limited distance upstream of the RC. We note that the retained upstream CBE-based *Cer/Sis* elements could also contribute to limiting RAG-scanning upstream from some secondary RCs established by inversional primary rearrangements. **d. One-loop-based scanning mechanism for $Ig\kappa$ secondary inversional rearrangements.** Our current findings indicate, in contrast to linear scanning in the *Igh*, inversional $V\kappa$ s, can join, at reduced frequency to deletional-oriented $V\kappa$ s during direct one-loop-based scanning through the secondary RC. While such utilization is mediated by strong $Ig\kappa$ RSSs, such inverted $V\kappa$ joining likely involves a short-range diffusional aspect for proper pairing. The mechanism for such a process remains to be elucidated. In theory, more distant $V\kappa$ elements such as CBEs might contribute.”

4.b. In addition, the authors are welcome to discuss the evolution of immune receptor loci in mammals.

Response:

Thanks for the suggestion.

We now added a short paragraph to the end of the legend of the Extended Data Fig.1 (line 814-820) stating: "**Evolutionary Implication of these findings.** Human $Ig\kappa$ and mouse $Ig\kappa$ have conserved organization and key regulatory elements, including *Cer/Sis*-like elements and predicted strong RSSs³, indicating potentially conserved $Ig\kappa$ two-loop-based primary and one-loop-based secondary rearrangements mechanisms. Also, such one-loop versus two-loop

joining mechanisms could also potentially contribute to other immune receptor loci that undergo just V to J rearrangements and have frequent editing, including Igλ in human and TCRα in human and mouse⁴⁸⁻⁵⁰.

5. Overall, the authors proposed a novel model of receptor editing, in which only a few immediate upstream Vks can be used for secondary recombination. Whether this mechanism plays a physiological role? For example, whether similar Vks are clustered and the receptor editing likely produces a successful H-L pair. The authors could discuss different hypotheses, although the validation is out of the scope of current manuscript.

Response:

We thank the referee for this insightful comment.

We have added the following sentence to the discussion (line 320-324): "*In this regard, Vks with high sequence similarity are semi-clustered across the Vκ locus⁴¹, which theoretically could promote fine-tuning of the Igκ repertoire by replacing autoreactive Vks with highly-related Vks harboring de novo generated CDR3 antigen-binding regions, which might dampen autoreactivity while promoting pairing with the existing IgH chain*".

6. There are some typos. For example, 3'RSS should be 3'RS in Line 183/240.

Response:

Thank you for the careful read. We have fixed these typos and several others.

Referee #2 (Remarks to the Author):

It is now well established that Igh locus V(D)J recombination occurs by deletion, whereas Iglk locus rearrangement involves either deletion or inversion. Igh locus VHDHJH rearrangement is mediated predominantly by RAG scanning whereas primary VkJk recombination is instructed by diffusion that Vk and Jk elements that are positioned into two distinct loop configurations. How secondary Vk-Jk rearrangements to remove self-reactive or non-functional VkJk rearrangements are orchestrated during the course of receptor revision has remained to be determined.

In this manuscript the authors show that deletional and inversional VkJk1 primary rearrangements, delete or displace a critical cis-regulatory elements, to generate pre-B cells that leaves the majority of Vk gene segments positioned in a deletional configuration upstream of the de novo (secondary) recombination center. The authors show that this remaining set of Vk genes undergo RAG scanning to generate a diverse set of VkJk rearrangements. In a parallel study the authors validate these findings by utilizing iPSC mouse models as well as Abelson lines that carry VkJk rearrangements to further boost support for a RAG scanning mechanism in the receptor revision process. Collectively, these data provide compelling evidence the receptor revision process involves a RAG-scanning mechanism.

How during the course of B cell progression, a tolerant and diverse immunoglobulin repertoire is assembled at the mechanistic level has remained an open question. In this study new and important insights are provided. The data are robust and compelling. The manuscript is also well written and results are presented in a well-organized fashion. I only

have a few comments that could further improve the manuscript.

Response:

We thank the referee for very positive comments about the quality of our study and the impact of our findings in this field. We also thank the referee for pointing out that: “The manuscript is also well written and results are presented in a well-organized fashion”.

Major comments:

1. Could the authors please include HiC maps to confirm expected differences in V κ -J κ chromosomal contacts for wild-type and Cer-Sis deleted pre-B cells?
2. It would be of interest to determine whether and how chromatin accessibility is altered across the recombination signal sequences in wild-type and Cis-Sis deleted pre-B cells.

Response

We combined the response to these two excellent comments and thank the referee for raising of these important points.

Both Comments:

Our current Cer/Sis-deleted mouse lines cannot be used for Hi-C or ATAC-seq experiments, as they need to be RAG-deficient and also complemented with a productive heavy chain allele to obtain pre-B cells in RAG-deficient mice to prevent V κ J κ rearrangements that would confound interpretation of results. This would take nearly one year. Thus, we have deleted Cer/Sis in WT and Cer/Sis-deleted RAG-deficient *v-Ab1* lines and used them for the suggested mechanistic studies (New Extended Data Fig.3; see below). Based on HTGTS-V(D)J-seq assays after complementation with RAG, both WT and Cer/Sis-deleted lines show a very similar rearrangement pattern as the corresponding mouse pre-B cells with or without Cer/Sis deletion (New Extended Data Fig.3b,e; see below). Based on this data, we used these lines for new studies as follows:

Comment 1:

We now routinely do Hi-C analyses in our lab; but still find, as we previously reported¹⁷, that while Hi-C assays give important information about broad chromosomal interaction across the genome, they still provide relatively low-resolution maps of interactions across *Igh*, *Ig κ* and most other chromosomal domains. In contrast, our rapid 3C-HTGTS assay has been proven to provide much higher resolution for analyzing local interactions across individual chromosome domains such as *Igh* or *Ig κ* ^{3,16,18,20}. Therefore, in response to this very good suggestion, we performed 3C-HTGTS baiting from the iE κ (RC) in WT and Cer/Sis-deleted RAG-deficient *v-Ab1* lines. These new studies reveal that J κ -RC interacts with upstream Sis and downstream enhancers in the WT *v-Ab1* line (New Extended Data Fig.3c and f; see below), consistent with the published interaction pattern in normal pre-B cells³. Upon Cer/Sis deletion, the J κ -RC interacts with upstream proximal V κ s, especially V κ 3-2 and V κ 3-7 which are also highly utilized upon Cer/Sis deletion (New Extended Data Fig.3c and f; see below).

Comment 2:

We thank the referee for raising yet another interesting point. We have now done the ATAC-seq in the *v-Ab1* lines outlined just above. Cer/Sis deletion did not markedly change the overall chromatin accessibility patterns across the V κ locus except for the very proximal V κ s just upstream of Cer/Sis (New Extended Data Fig.3d and g; see below). Indeed, the proximal V κ s that become more accessible are those, such as V κ 3-2 and V κ 3-7, that are much more highly utilized upon Cer/Sis deletion (New Extended Data Fig.3; see below).

These findings correlate with a report²⁴ that Cer/Sis deletion activates transcription of proximal V κ s. We have now cited this paper for this finding. Consistent with this, in the V κ /J κ 1-pre-rearranged line with Cer/Sis displacement, transcription of RC-proximal V κ s is well correlated with and required for interaction with the secondary RC and their rearrangements (Fig. 4). Of note, the ATAC-seq now shows that distal and middle V κ s are accessible but not utilized upon Cer/Sis deletion. We now noted in the text that this finding further supports our conclusion that saturated rearrangement of secondary RC-proximal upstream V κ s via linear RAG scanning limit usage of more distal V κ s. We have also added it to the text (line127-130): "*Together, these results indicate both transcription-mediated increased RC interactions¹⁷ and V κ -RSS chromatin accessibility²⁵⁻²⁹ may synergistically contribute to the high usage of proximal V κ s during linear RAG scanning upon Cer/Sis deletion*", which also addresses a comment made by reviewer 3

Combined response to comments 1 and 2 in main text (lines117-130)

" To further investigate the mechanistic basis of the dominant usage of V κ 3-2 and V κ 3-7 upon Cer/Sis deletion, we performed HTGTS-V(D)J-seq and 3C-HTGTS¹⁶ in WT and Cer/Sis-deleted v-Abl lines. HTGTS-V(D)J-seq results in RAG-sufficient WT and Cer/Sis-deleted v-Abl lines were very similar to those in pre-B cells (Extended Data Fig.3a,b,e). Upon Cer/Sis deletion, V κ 3-2 and V κ 3-7 were dominantly utilized, and also prominently interacted with the RC, as revealed by 3C-HTGTS in RAG-deficient WT and Cer/Sis-deleted v-Abl lines (Extended Data Fig.3c,f), likely due to enhanced proximal V κ transcription^{17,24}. In addition, ATAC-seq in these RAG-deficient lines revealed chromatin accessibility of these two highly utilized proximal V κ s substantially increased, while upstream V κ s, which were not used, maintained their varying levels of accessibility (Extended Data Fig.3d,g). Together, these results indicate both transcription-mediated increased RC interactions¹⁷ and V κ -RSS chromatin accessibility²⁵⁻²⁹ may synergistically contribute to the high usage of proximal V κ s during linear RAG scanning upon Cer/Sis deletion. "

Minor comment:

References should perhaps be a bit more balanced and include those of others that also have contributed to this field.

Response:

We apologize for this. We have now included 18 additional related references for others at relevant places in the revised manuscript and highlighted them in Red in the revision to facilitate identification.

Extended Data Fig. 3 (Related to Fig.1)

Referee #3 (Remarks to the Author):

The manuscript by Li et al presents a convincing mechanism by which gene segments are brought into close proximity for secondary recombination in the mouse immunoglobulin kappa locus (Igk). The Igk locus is unusual in that it undergoes both deletional and inversional recombination. The authors previously presented a two-loop extrusion model for primary Igk recombination whereby cohesion-mediated loop extrusion allows RAGs bound to Jk1 to linearly scan upstream Vk gene segments but only as far as the Cer/Sis element. Recombination to more distal Vk RSSs required requires a second loop through which Vk RSSs can encounter the Jk1-bound RAGs via short-range diffusion. Here, the authors examined how secondary recombination occurs to Jk2, Jk4 and Jk5 using high throughput sequencing and gene editing approaches. They convincingly show that following primary recombination events that either delete the Cer/Sis element or recombine it such that it is distal to the originally inversional middle Vk RSSs, RAGs are able to linearly scan upstream Vk RSSs in a one-loop extrusion model, including middle RSSs that now undergo deletional recombination. There are some exceptions where RSSs in the inversional orientation lie next to the recombination centre and in these cases, the authors propose a short-range diffusion model.

The experiments are carefully performed and well presented. Given the high frequency of secondary recombination, the data explain an important and fundamental step in the generation of the antigen receptor repertoire. As such, they are likely to be of broad interest to people in the V(D)J recombination field and have implications for long range chromatin interactions that will be of interest to colleagues studying higher order chromatin folding/interactions. The outstanding features are the neat use of iPSC and well controlled gene editing experiments.

Although this is an extremely interesting and well presented manuscript, there are some points that should be addressed.

Response:

We thank this referee for the accurate summary of the key points and conclusions presented in our manuscript. We greatly appreciate the referee's highly favorable evaluation of the quality of our studies and the significance of our conclusions in the field and also the broader impact of our findings.

Major points:

1) The authors neatly show how secondary recombination to Jk2, Jk4, and Jk5 can occur. However, there are also substantial amounts of secondary rearrangement to the RS element (Yamagami et al., 1999). This should be discussed.

Response:

We thank the referee for this important comment. We fully agree that in addition to J_κs, the downstream 3'RS in mouse or KDE in human mediate substantial amounts of secondary (editing) rearrangements, particularly critical for κ/λ light chain isotype exclusion. Based on single cell analyzes in mouse B cells (Yamagami et al., 1999a)⁴, about 33% of small pre-BII cells contain at least one V_κ/3'RS rearrangement. In addition, the 3'RS also efficiently joins to the iRS (intronic recombining sequence between J_κ5 and C_κ segment) at

about half of the frequency of $V_{\kappa}/3'RS$ rearrangements (Yamagami et al., 1999b)³⁴. We have referenced several relevant published papers and mentioned the role of 3'RS-mediated secondary rearrangements in the text (line 201-204): “The downstream 3'RS has been reported to efficiently join to $V_{\kappa}S$ and iRS (intronic recombining sequence between $J_{\kappa}5$ and C_{κ}), which substantially contributes to Ig_{κ} secondary rearrangements and is particularly critical for κ/λ light chain isotype exclusion^{4,10,33,34}”

In response to this comment, we also present here preliminary HTGTS-V(D)J-seq using a 3'RS bait in our ν -*Abl* line that indeed confirms abundant joining of the 3'RS both to the intronic iRS (30%) and $V_{\kappa}S$ (70%) (Preliminary Figure on the right that is not shown in the manuscript). These results indicate that our assays will be suitable for more in-depth studies of how the mechanism and regulation of the 3'RS-mediated secondary rearrangement going forward. Further addressing this interesting question requires significant modifications of test systems and is a long-term goal of our lab.

2) To highlight the importance of secondary recombination to the Ig_{κ} repertoire, it would be very helpful if the authors cite the levels of secondary recombination that have been determined such as in the single cell analyses by Yamagami et al (1999).

Response:

Thanks for pointing this out. We now cite this paper in the abstract when mentioning the importance of secondary recombination to the Ig_{κ} repertoire (line 39-41): “Secondary Ig_{κ} rearrangements replace non-functional or autoreactive primary $V_{\kappa}J_{\kappa}1$ rearrangements, expanding the Ig_{κ} repertoire and mediating central tolerance via receptor editing^{4,6-11}”. We have now also mentioned in the introduction (line 84-87): “While secondary V_{κ} rearrangements to $J_{\kappa}2$, 4, and 5 generate a large proportion (approximately 40-60% based on single cell analysis in mouse B cells) of the Ig_{κ} repertoire^{4,8-10}, the mechanism of this physiologically critical process has remained speculative³”.

3) Could the authors please clarify why, in Fig 1g, scanning from $J_{\kappa}5$ only extends for a limited distance into the middle V_{κ} s upon deletion of the Cer/Sis element whereas in Fig 2f, scanning from $J_{\kappa}5$ extends across the locus?

Response:

Thanks for this question. In the previous Fig. 1g (now Fig. 1j in the revised manuscript), scanning is assayed in a Cer/Sis-deleted line and represents scanning from an ectopic $J_{\kappa}1$ -RC which extends a limited distance upstream due to saturation of rearrangements by the most proximal V_{κ} s. These rearrangements set-up secondary RCs only in this most proximal V_{κ} region. Thus, secondary rearrangements from $J_{\kappa}5$ linearly extend from the proximal $V_{\kappa}/J_{\kappa}1$ -RCs only for a limited distance upstream before being saturated, as is shown in Fig. 1i-k. In the previous Fig. 2f (now Fig. 2g in the revised manuscript), scanning occurs from 100 plus primary V_{κ} - $J_{\kappa}1$ RCs generated across the V_{κ} locus by primary $J_{\kappa}1$ - V_{κ} rearrangements via the two-loop diffusional mechanism. Thus,

limited linear scanning from the numerous secondary RCs across the locus collectively covers the entire $V\kappa$ locus.

We have clarified this point in the text (line 154-156) as follows: "These findings indicate that limited linear scanning from the numerous secondary $V\kappa J\kappa 1$ -based RCs established during primary joining collectively covers the entire locus. (Extended Data Figure 4 and legend)". To further clarify this point, we have included a new Extended Data Figure 2 and illustrate $J\kappa 1$ - and $J\kappa 5$ scanning in absence of Cer/Sis in the legend (line

822-835) as follows: "Extended Data Fig.2 Illustration of $V\kappa$ - $J\kappa$ rearrangements in Cer/Sis-deleted BM pre-B cells. Related to Fig.1. a. Illustration of WT $Ig\kappa$ locus (not to scale). b. Illustration of Cer/Sis-deleted $Ig\kappa$ locus that Cer/Sis is deleted by CRISPR/Cas9-mediated editing to creates an ectopic RC for $J\kappa 1$ -based primary rearrangement. c. Illustration of production of pre-rearranged $V\kappa 3-2/J\kappa 1$ line, in which $J\kappa 1$ joins to a proximal deletional $V\kappa 3-2$ via linear RAG scanning from the ectopic primary $J\kappa 1$ -RC. This rearrangement also creates a new $V\kappa 3-2/J\kappa 1$ -RC for secondary rearrangements. d. Illustration of production of pre-rearranged $V\kappa$ - $J\kappa 5$ line, in which $J\kappa 5$ joins to a proximal deletional $V\kappa 3-7$ via linear RAG scanning from the secondary $V\kappa 3-2/J\kappa 1$ -RC. These illustrations show an example that, in Cer/Sis-deleted BM pre-B cells, RAG scanning from an ectopic primary $J\kappa 1$ -RC is limited to the proximal $V\kappa$ locus and creates $V\kappa/J\kappa 1$ secondary RCs that are also restricted in the proximal $V\kappa$ locus. Thus, secondary rearrangements from other $J\kappa$ s, including $J\kappa 5$, linearly extend from the proximal $V\kappa/J\kappa 1$ -RCs only for a limited distance upstream before being saturation."

4) Figure 4 and lines 206-207. The authors claim that transcription is an impediment to linear scanning. However, it is the highly transcribed Vks that undergo most rearrangement (Vk6-32, Vk8-34 and Vk18-36) and removal of the Vk18-36 promoter substantially reduces its recombination. Could this also suggest that transcription is NEEDED for recombination? What is the evidence that transcription is an impediment to linear scanning? This must be addressed.

Response:

Thanks for pointing this out. We have previously shown that transcription can impede loop-extrusion-mediated RAG scanning and promote rearrangements of paired RSSs within the impeded region¹⁷. We have shown a similar role for transcriptionally impeded loop extrusion in CSR⁴⁴. The role of transcription in impeding loop extrusion is now widely accepted^{17,44-46} and this is now referenced in the Extended Data Figure 1 Legend (see response to reviewer 1). In the current study, we similarly show in Fig. 4 that deleting the promoter of $V\kappa 18-36$ largely abrogates its transcription, interaction with the

Extended Data Fig. 2 (Related to Fig.1)

secondary RC and robust utilization. Importantly, V κ 18-36 has a weak RSS, but replacing the weak RSS with a strong RSS allows it to rearrange even without measurable RC interaction (new Fig. 4c,iv). However, we agree that these data also are consistent with reduced transcription impacting transcription-mediated RSS chromatin accessibility. We have now again made this important point, also raised by reviewer 2 (see above), in the text (line 236-239) for Fig.4 as follows: “*These findings indicate that utilization of V κ s with weaker RSSs may be directly enhanced by associated transcriptional impediments that increase their association with the secondary RC and mediate RSS chromatin accessibility during the linear RAG scanning-based Ig κ secondary recombination*”.

5) Extended Data Figure 4c: Can the authors explain why recombination to V κ 17-121 is almost completely lost in the inverted orientation? Conversely, why does recombination to V κ 11-125 show a greater increase relative to the other V κ s analysed in the inverted orientation?

Response:

We thank the referee for raising this interesting point. The explanation is directly related to the relative RSS strength of these two V κ s. V κ 17-121 has a relatively weak RSS (RIC:-18.59, Supplementary Table 1). Thus, it could be highly used for secondary deletional rearrangements via linear RAG scanning, but not diffusion-based secondary inversional rearrangements that highly rely on RSS strength³. In contrast, V κ 11-125 has a very strong RSS (RIC:-10.84, Supplementary Table 1). Thus, it could be highly used when placed proximal to secondary RC via diffusion-based secondary inversional rearrangements. We have added this important point to the legend of Extended Data Figure 6c (line 959-963) as follows: “*Of note, V κ 11-125 with a strong RSS (Supplementary Table 1), but not V κ 17-121 with a weak RSS (Supplementary Table 1), was robustly utilized for secondary inversional rearrangements during RAG scanning, indicating a strong RSS is required for such an inversional rearrangement process*”.

6) It would be helpful to include the significance of various changes in recombination levels in the figure legends as this group has done in previous publications. For example:

- Line 170: “edited chimeras were highly enriched for productive V κ s/J κ 4 rearrangements” Figs 3c,f. The significance of the increase in productive rearrangements would be useful.
- Line 209: “which markedly reduced V κ 18-36 rearrangement” Fig 4biii. The significance of the decrease would be useful.
- Line 242: “deletional-oriented V κ 6-17 is the most robustly rearranged; but, at least three inversional V κ s, including V κ 8-24 about 300kb upstream, are significantly utilized” Please give the significance and what it is being compared to.

Response:

Thanks for pointing this out. We have now added the significances to the legends and clarified what being compared in the legends as follows:

In Fig.3 Legend: “*The percentages of productive rearrangements in edited chimeras compared to those in parental chimeras are significantly increased (P value=0.0011).*”

In Fig.4b Legend: “*Absolute utilization of V κ 18-36 is significantly decreased upon its promoter deletion (P value<0.0001).*”

In Extended Data Fig.8b Legend: “Absolute utilization of V κ 6-17 is higher than that of the second highly utilized V κ 8-19 (P value=0.001).”

In Extended Data Fig.8b Legend: “Absolute utilization of three inversional V κ s, V κ 8-19, V κ 8-21 and V κ 8-24, are higher than that of the fourth highly utilized inversional V κ 6-32 (P value=0.015 for V κ 8-19, P value=0.0002 for V κ 8-21, P value=0.0046 for V κ 8-24).”

Minor points:

1) Fig 5a labelling: “inverion” should read “inversion”.

Response:

Thanks for careful read. We have fixed this typo.

2) It would be helpful to refer to the RS element as such, rather than 3' RSS. As written, this is confusing.

Response:

We apologize for this. We now appropriately refer to the 3'RS element in the text.

3) Line 121: The authors claim RAG scanning from the primary J κ 1-RC is terminated at Sis. However, Fig. 1c reports 219 junctions beyond Sis. Could the authors please add more explanation, perhaps based on the levels of scanning beyond Sis. It is hard to compare with the levels in Fig. 2f as the scales on the y axes are different.

Response:

Thanks very much for pointing this out. We agree that the most likely explanation for these extremely low-level upstream junctions could be low-level RAG scanning beyond Cer/Sis, based on the predominantly deletional orientation pattern of these low-level junctions. We could show a blow-up of this portion of Fig. 1c in an Extended Data figure with a broken y axes for the junctions downstream of Sis to be able visualize the very low-level junctions, if the reviewer thinks it is needed. However, confirming any mechanism, for such very low levels would require additional experimentation, and, whatever the mechanism, it would not influence any of our conclusions. Currently, to address this minor comment, we mention in the revised text (line 99-101): “In normal pre-B cells, primary J κ 1 joins utilize V κ s across the locus in deletional and inversional orientation, while RAG scanning is “largely” terminated at Sis (Fig.1a-d)”.

4) Supplementary Table 2 contains the coding sequences of the V-J joins but it is not clear which sides of the sequence as well as the breakpoint which makes it hard to interpret. These could be highlighted to allow for better understanding. Further, there appears to be no sequence tracks for any of the following derivative lines:

- the pre-rearranged V κ 5-37/J κ 1 v-Abl cell line to generate its V κ 18-36 promoter-deleted derivative;
- the pre-rearranged V κ 1-117/J κ 1 v-Abl cell line to generate a V κ s-inverted derivative
- the pre-rearranged V κ 3-12/J κ 1 v-Abl cell line to delete the DNA segment from upstream of J κ 3 through downstream of J κ 5;
- the pre-rearranged V κ 3-12/J κ 1 v-Abl cell line to generate its J κ 2-inverted derivative. All candidate clones with desired genetic modifications were confirmed by Sanger sequencing (line 467)

Response:

Thanks for pointing this out. We have now included the detailed sequences for all the cell lines used in this study and labeled them with different colors to label the sequences and breakpoint in the revised Supplementary Table 2.

Comments on Code:

I am not an expert to review the code but all code seems accessible from the links provided. However, the method of Gro-Seq analysis was not given in the Code Availability statement. Please can this be added?

Response:

Thanks for pointing this out. We have now revised the code availability in the revised manuscript as follows: “*The pipeline for GRO-seq data analysis is available at https://github.com/Yyx2626/Fred_Alt_Lab/tree/master/GROseq. The pipeline for ATAC-seq data analysis is available at <https://github.com/nf-core/atacseq>. The mouse Igk-specific cryptic RSS usage analysis pipeline is available at https://github.com/Yyx2626/HTGTS_related/tree/main/Igk_specific_anno_and_filter.”*

References in the responses to all reviewers are listed based on the numbers of the citations as listed in the text)